

# Combining airborne in situ and ground-based lidar measurements for attribution of aerosol layers

Anna Nikandrova[1], Ksenia Tabakova[1], Antti Manninen[1], Riikka Väänänen[1], Tuukka Petäjä[1], Markku Kulmala[1], Veli-Matti Kerminen[1] and Ewan O'Connor[2]

[1]University of Helsinki, Department of Physics P.O. Box 64, FIN-00014, Finland

[2]Finnish Meteorological Institute, Helsinki, Finland

*Correspondence to*: Anna Nikandrova (anna.nikandrova@helsinki.fi)

**Abstract.** Understanding the distribution of aerosol layers is important for determining long range transport and aerosol radiative forcing. In this study we combine airborne in situ measurements of aerosol with data obtained by a ground-based High Spectral Resolution Lidar (HSRL) and radiosonde profiles to investigate the temporal and vertical variability of aerosol properties in the lower troposphere. The HSRL was deployed in Hyytiälä, Southern Finland, from January to September 2014 as a part of the US DoE ARM (Atmospheric Radiation Measurement) mobile facility during the BAECC (Biogenic Aerosols

– Effects on Cloud and Climate) Campaign. Two flight campaigns took place in April and August 2014 with instruments measuring the aerosol size distribution from 10 nm to 10 μm at altitudes up to 3800 m. Two case studies from the flight campaigns, when several aerosol layers were identified, were selected for further investigation: one clear sky case, and one partly cloudy case. During the clear sky case, turbulent mixing ensured low temporal and spatial variability in the measured aerosol size distribution in the boundary layer whereas mixing was not as homogeneous in the boundary layer during the partly

cloudy case. The elevated layers exhibited greater temporal and spatial variability in aerosol size distribution, indicating a lack of mixing. New particle formation was observed in the boundary layer during the clear sky case, and nucleation mode particles were also seen in the elevated layers that were not mixing with the boundary layer. Interpreting local measurements of elevated layers in terms of long-range transport can be achieved using back trajectories from Lagrangian models, but care should be taken in selecting appropriate arrival heights, since the modelled and observed layer heights did not always coincide. We

conclude that higher confidence in attributing elevated aerosol layers with their air mass origin is attained when back trajectories are combined with lidar and radiosonde profiles.

## 1 Introduction

Aerosols are tiny particles suspended in the atmosphere that severely affect human health (Tie et al, 2009; Apte et al., 2015; Pope et al., 2015) and climate. Most of the particles have a direct cooling effect on climate by scattering solar radiation

(McCormick and Ludwig, 1967; Sundström et al., 2015, Lacagnina et al., 2017) and indirect by changing cloud properties



(Haywood and Boucher, 2000; Ten Hoeve and Augustine, 2016; Saponaro et al., 2017), yet some of the particles have a warming effect by absorbing solar radiation (Yu et al., 2006). The average lifetime of aerosol particles in the boundary layer (BL) vary from several hours to two weeks (Seinfield and Pandis, 2006) and they can be transported far from their source of origin. However, they are not distributed uniformly, with aerosol concentrations and properties varying significantly in space and time. Thus, it is challenging to implement aerosol schemes in global climate models (Myhre et al., 2013; Zhang et al., 2016; Glassmeier et al., 2017) and the impact of aerosol remains one of the largest sources of uncertainty in climate predictions (IPCC, 2013).

Most aerosol particles are emitted from the surface (Kaufman et al., 2002), or are formed from their pre-cursor gases either close to the surface or higher up in the free troposphere (e.g. Kulmala et al, 2004, 2007, 2013; Dunne et al., 2016). Turbulent mixing distributes aerosol uniformly throughout the well-mixed BL, but stable stratification elsewhere in the atmosphere usually inhibits any further mixing. Hence, layers can form once the source of turbulent mixing is removed, one example being the formation of residual layers after sunset (Stull, 2012). Here, we define elevated layers as those existing above the daytime well-mixed BL. This can include residual layers if the boundary-layer depth is supressed on subsequent days. Deep convection (Andreae et al., 2001) and mid-latitude cyclones (Sinclair et al., 2010) can transport aerosol vertically throughout the troposphere, and, once in the free troposphere, baroclinic systems can advect aerosol over long distances (Donnell et al., 2001). Wet deposition and evaporation to the gas phase are the main removal mechanisms for aerosol above the BL, although it is also possible for elevated aerosol layers to be mixed back into the BL after some time of being aloft.

The vertical distribution of aerosol particles is important for determining the direct and indirect aerosol radiative forcing (Haywood and Ramaswamy, 1998). Lidar measurements are able to track the evolution of aerosol layers with a high resolution in space and time (Wandinger et al., 2002; Groß et al., 2011; Burton et al., 2012; Pappalardo et al., 2014; Baars et al., 2016). Reid et al. (2017) looked at the monthly variability of backscatter profiles from a High Spectral Resolution Lidar (HSRL) located in Hunstsville, Alabama, US during summer and reported that aerosol backscatter was the highest below 1.5 km and decreasing with an increasing height rapidly until 3.5 km. They also observed occasionally different layering structures in the free troposphere, which was in general a clear region with low aerosol concentration. During the Two-Column Aerosol Project (TCAP, Berg et al., 2016), studying the atmospheric column both at the coast of North America and several hundred kilometres away in the Atlantic Ocean, elevated aerosol layers were observed on four out of six clear-sky research flights with contributions of up to 60 % to the total column aerosol optical depth. Fast et al. (2016) found that some of these elevated aerosol layers were likely lifted from the BL as a result of strong synoptic-scale convergence. At higher latitudes, smoke events may be responsible for elevated layers with significant contributions to the total column aerosol optical depth (O'Neill et al., 2008).



An airborne HSRL-2 was used to constrain the vertical distribution of aerosol microphysical properties observed in California and Texas (Sawamura et al., 2017). Microphysical properties retrieved from HSRL-2 showed a good agreement with in situ measurements; however, calculated backscatter and extinction coefficients were consistently underestimated, which was attributed to the undersampling of coarse mode particles by in situ measurements. Combined data from diverse measurement campaigns over the Pacific show that the free troposphere was dominated by aerosols formed near cloud edges and in convective regions, as well as particles transported from continents (Clarke and Kapustin, 2002).

Detailed information on aerosol size distributions and microphysical properties can be obtained from in situ airborne measurements. However, compared to the quantity of aerosol measurements at the surface, there have been relatively few flight campaigns investigating elevated aerosol layers, especially at low aerosol load conditions. Boy et al. (2004), O'Dowd et al. (2009) and Schobesberger et al. (2013) conducted airborne measurements over a boreal forest, primarily interested in new particle formation (NPF). New particles were observed throughout the BL in all three studies, but Schobesberger et al. (2013) reported much lower particle concentrations outside the BL. This suggests that in the boreal forest large-scale NPF events are typically confined to the BL, as only one event was detected in the free troposphere.

In this study, our aim was to investigate aerosol layers in a rural environment, their origin, and how they change over time. For this purpose, a comprehensive set of ground-based remote sensing observations together with both airborne and ground-based aerosol measurements were collected during the Biogenic Aerosols – Effects on Cloud and Climate (BAECC) campaign in Hyytiälä, Finland during 2014 (Petäjä et al, 2016). We used HSRL measurements from the surface, and Scanning Mobility Particle Sizer (SMPS) and Optical Particle Counter (OPC) measurements on board an aircraft, described in Section 2, to analyse aerosol layers in two case studies, described in Section 3. The first case represents typical clear-sky weather conditions during spring at the station with a clean air mass arriving from the north. This is an ideal case because the development of the BL and elevated aerosol layers could be monitored for several days without interruption. The second case is more complicated, with partly cloudy and unstable atmospheric conditions.

## 2 Experimental setup

The SMEAR-II (Station for Measuring Forest Ecosystem-Atmosphere Relations - II, see Hari and Kulmala, 2005) measurement station located in Hyytiälä, southern Finland (61 51 N, 24 17 E, 181 m a.s.l.), is a rural background station with no major anthropogenic emission sources located nearby. During the BAECC campaign, the US Department of Energy Atmospheric Radiation Measurement (ARM) programme deployed the HSRL in Hyytiälä from January to September 2014 as a part of the ARM mobile facility (AMF). Vaisala RS92 radiosondes (RS) were launched 4 times a day during the campaign (nominally at 00Z, 06Z, 12Z and 18Z).



## 2.1. Instrumentation

**High Spectral Resolution Lidar**

The AMF HSRL (Shipley et al., 1983; Grund and Eloranta, 2005; She et al., 1992) is an autonomous lidar system designed to retrieve vertical profiles of the backscatter coefficient, extinction coefficient, and depolarisation. The system uses a frequency-doubled Nd:YAG laser emitting pulses at a wavelength of 532 nm and a repetition rate of 4 kHz, together with an afocal telescope with a diameter of 40 cm acting as both transmitter and receiver. The telescope has a field of view of 45 µrad, which limits the impact of multiple scattering, and the large expansion of the outgoing beam means that the system is eye-safe, permitting the flight campaign to operate in the immediate vicinity of the instrument. The emitted laser light is circular-polarised. The detection chain utilises photon counting to record the atmospheric return in three channels: combined, molecular, and cross-polarisation, at 0.5 seconds and 7.5 m resolution. The combined channel contains backscattering from both particulates and molecules, whereas the molecular channel includes an iodine absorption filter (Eloranta and Razenkov, 2006) in the path to record molecular scattering only, and the cross-polarisation channel measures the degree of circular depolarisation. Full details on the instrument setup are available in the ARM HSRL instrument handbook (Goldmsith, 2016). The profile of attenuation is determined from the known profile of molecular scattering, enabling direct retrievals of extinction, backscatter, and particulate depolarisation up to an optical depth of 4. To reduce noise, the raw data was averaged to 5 s and 30 m before deriving the backscatter, extinction, and circular depolarisation profiles.

**Airborne aerosol measurements**

In situ airborne data in the lower atmosphere were obtained with a Cessna 172 light aircraft, modified for the research flights by replacing the backseats with a rack for the instruments (see. Schobesberger et al., 2013, and Väänänen et al., 2017). The sample air was collected from under one wing, away from the engine exhaust, and transferred inside the cabin via a stainless tube (inner diameter 22 mm, length 4.2 m). The inlet line was a downscaled version of one used by the University of Hawaii DC-8 (McNaughton et al., 2007), and the flow of the main inlet line was kept at 50 l min$^{-1}$.

The aerosol and gas instruments were situated in a rack inside the cabin. The total aerosol number concentration was measured with an ultrafine Condensation Particle Counter (uCPC, Model 3776, TSI Inc.), whereas a Scanning Mobility Particle Sizer (SMPS) was used to determine the particle number size distribution in the diameter range of 10-230 nm with a temporal resolution of two minutes. The SMPS comprised a short Hauke type Differential Mobility Analyser (DMA) with closed-loop sheath air, and a TSI 3010 CPC as a particle counter. SMPS data were inverted using the method introduced by Collins et al (2002). An Optical Particle Sizer (OPS, Model 3330, TSI Inc.) measured the particle number size distribution in the diameter range of 300-9000 nm with a temporal resolution of 10 s. Additionally, a relative humidity sensor was installed under the inlet. All particle and gas instruments were calibrated in the laboratory prior to the campaign, with errors in CPC total concentrations



and SMPS particle counts below 10 %. All aerosol data were corrected to the standard temperature and pressure (100 kPa and 273.15 K).

One intensive flight campaigns took place in spring and another one in autumn, 2014. A typical measurement flight took 2-3 hours and consisted of numerous legs of about 40 km in length flown above the SMEAR II measurement station at Hyytiälä. The Cessna 172 air speed was low, around 35 m s$^{-1}$ (130 km h$^{-1}$), enabling a relatively high data resolution (4.2 km for SMPS and 350 m for OPS). The maximum flight ceiling was 3800 m a.s.l. A typical flight plan consisted of a climb up to the free troposphere, and constant altitude legs at different altitudes. A typical climb or descent rate during the flights was 2.5 m s$^{-1}$. A GPS instrument was used to record the flight track.

## 2.2. Methods

We investigated aerosol size distributions in the BL and elevated atmospheric layers that were identified by utilizing HSRL backscatter and depolarization fields. A wavelet decomposition was used to determine layer boundaries, similar to the approach used in STRAT (Morille et al., 2007). Since our work is based on individual case studies, suitable coefficients for the algorithm were decided based on a visual inspection. On a clear sky day, most of the aerosol load in the boreal forest area is concentrated in the BL. Therefore, the BL was easily distinguishable by the high peaks in backscatter coefficient as a consequence of a strong aerosol scattering. Layers classified with the HSRL were confirmed with the RS measurements as areas of changes in specific and relative humidity profiles most often indicate edges of layers. The closest in-time RS measurements were used if the time of Cessna flight and the RS launch time did not match. We considered in this study only layers below 3800 m, as it was the maximum altitude of Cessna.

The SMPS and OPS measurements were combined in order to obtain a size distribution ranging from 10 nm to 10 μm. Because the SMPS measures the dry size (particles are dried prior to entry), and OPS measures the ambient size (at an ambient relative humidity), the measured SMPS size distributions were modified to represent the growth expected at the ambient relative humidity by using a growth factor (GF) calculated according to Laakso et al. (2004). The ambient size distributions were then grouped using the layer boundaries found in the lidar data. Only data in the vicinity of Hyytiälä were used, with data obtained within 50 km of the airport excluded from any analysis.

Here, we will use the term nucleation mode for particles smaller than 25 nm, Aitken mode for particles ranging from 25 to 100 nm, accumulation mode for particles from 100 nm to 1 μm, and supermicron particles for particles from 1 to 10 μm. All times are given in the Eastern European Summer Time (UTC+3).

To look at the spatial variability of arriving air masses in space and height, we calculated 96-hour back trajectories for every 50 m for the altitudes from 500 to 3500 m AGL using the HYSPLIT model (Stein et al., 2015).



## 3 Results and Discussion

Two case studies were selected, both with air masses coming from the north but with different atmospheric conditions. The first case study (Case I) consists of 3 sequential clear-sky days with NPF events detected at the ground level each day. The second case study (Case II) consists of a single day with low-level clouds present and no NPF taking place.

### 3.1. Case I: typical clear sky situation during 8 - 10 April

Figure 1a displays the HSRL backscatter coefficient from 50 m to 4000 m for Case I, with higher values of backscatter cross section indicating either higher particle concentrations or larger particle sizes. The figure illustrates how the BL and other layers were developing, evolving and mixing during this period. The amount of lidar depolarization, shown in Fig. 1b, depends on the particle shape, and also clearly illustrates the evolving atmospheric structures and their boundaries. Sharp changes in the relative humidity (RH) seen in the RS profiles also agree with the layer determination obtained from the HSRL backscatter cross section and depolarization. However, not all HSRL determined layers exhibit a corresponding change in RH (for example, during 8 April at 14:00 at 2800 m).

The simplified layer structure shown in Fig. 1c was obtained from the HSRL backscatter and depolarization fields. Four layers were identified, denoted as the first, middle, upper and high layers. The first layer includes both the BL and the residual layer as it was not possible to separate them with our simple algorithm. Figure 1c demonstrates that, in April, the first layer in Hyytiälä can reach up to 1500 m during the day, and is usually shallow at night (lower than 1000 m). There were several elevated layers on 8 April, one of which disappeared during the day, with no major new layers appearing on 9 April. On 10 April, a new layer exhibiting high backscattering and low depolarisation developed at around 2500 m. Additionally, a narrow band with a high backscatter cross section (relative to surroundings) during 9 and 10 April was initially classified as a separate layer and then termed an interface zone after a closer inspection. During these three days, six flights were made with the Cessna (one morning flight and one afternoon flight each day). Four flights were selected for analysis; these are described in more detail below and summarised in Table 1. In figure 2, for each flight, a panel comprising three plots displays 1) radiosonde profiles of RH and specific humidity mixing ratio; 2) time-height HSRL backscatter cross section with the Cessna flight altitude superimposed; and 3) mean and +/- standard deviation of the aerosol size distribution obtained from SMPS and OPS measurements within each layer.

### 3.1.1. Case I: Flight descriptions

Four layers in the HSRL backscatter coefficient were recognized during the Cessna flight between 16:00 and 17:30 EEST on 8 April 2014, as shown in Fig. 2b. The RH and specific humidity mixing ratio from the RS (Fig. 2a), launched two hours before the flight at 14:22 EEST, also shows several distinct layers in the profile which correspond well with those identified





by the HSRL. The profile of the specific humidity mixing ratio indicated that this layer was well mixed, whereas the corresponding values were changing through the other layers.

The size distributions displayed four distinct distribution shapes corresponding to the four layers observed by the HSRL (Fig. 2c). The BL was characterized by high aerosol concentrations of up to 6000 cm$^{-3}$ in the Aitken mode, due to the NPF event that took place earlier in the afternoon. The lowest concentrations of Aitken mode particles were found in the first middle layer. The second middle layer had a similar size distribution shape for particles smaller than 100 nm but higher concentrations, and displayed the highest concentrations of supermicron particles, even higher than in the BL. The second middle layer also exhibited much more depolarisation than the other layers, together implying long-range transport of large non-spherical particles. Nucleation mode particle concentrations were higher than Aitken mode particle concentrations in both middle layers, whereas no particles smaller than 15 nm were detected in the upper layer. However, the upper layer had much higher concentrations of 20-40 nm particles than the other elevated layers.

The first Cessna flight on 9 April took place from 11:00 to 12:30 EEST (Fig. 2d-f). Three distinct layers were observed below 3500 m, also visible in the RS humidity profiles. The middle layer was significantly drier than both the BL and the layer above. The mean aerosol size distribution in the BL is shown separately for the ascent and descent profiles (Fig. 2f), illustrating that there was a notable increase in the nucleation mode particle concentration during the descent. The middle layer was characterized by a low Aitken mode concentration and high accumulation and supermicron mode concentrations. The upper layer displayed a similar aerosol size distribution shape to the one in the BL, but with considerably smaller concentrations.

The same three layers were seen during the second Cessna flight, which took place from 16:00 until 17:40 EEST (Fig. 2g-i). The humidity profiles were also similar to the morning flight. The impact of the NPF and subsequent growth is clearly seen in the BL aerosol size distribution (Fig. 2i), with Aitken mode concentrations reaching 5000 cm$^{-3}$, much higher than observed during the morning. The size distributions in the middle and upper layer were similar to the morning flight except for the nucleation mode. No particles smaller than 15 nm were detected in the upper and middle layers during the morning flight, but these were observed in the afternoon flight, providing evidence for NPF in elevated layers.

On 10 April the Cessna flew in the afternoon from 13:45 to 15:30 EEST (Fig. 2j-l). Three distinct layers are visible in the humidity profiles, but the HSRL data suggests four, subdividing the upper layer into two. In addition, the HSRL observed a very thin layer which we will discuss separately in section 3.1.5. The aerosol size distributions in each layer are similar to those in the previous day. The upper layer also exhibited higher concentrations in the supermicron mode.

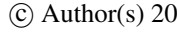



### 3.1.2. Case I: Size distribution variability within layers

We also investigated the variability in the aerosol size distribution within each layer, which is illustrated in Fig. 2c,f,i,l with the standard deviation above and below the mean size distribution for each layer. The least variability was usually observed in the BL, indicating that this layer was well-mixed vertically and horizontally with similar aerosol concentrations at all heights. Large variability was sometimes seen in the ultrafine range in the BL, which was attributed to NPF events. NPF events took place every day during the case study period. For the morning flight on 9 April, the variability for all size ranges was low during the ascent but increased in the ultrafine range during the descent (Fig. 2f); the ascent and descent profiles were separated by an hour and the NPF event began after the Cessna left the BL while ascending. During the afternoon flight on the same day 4 hours later, when the NPF event had finished, the BL appeared homogeneous again with a low variability for almost all sizes except for particles between 20 and 30 nm (Fig. 2i). On 10 April, the variability in the BL nucleation mode was also quite high, a result of the NPF event still ongoing (Fig. 2l).

In contrast, a higher variability in the size distribution was seen for the elevated layers, where there is much less turbulent mixing, with more variation seen at almost all size ranges. The middle layer exhibited a similar variability across all three days, whereas the upper layer showed some changes from day to day, probably due to changes in the depth of the layer. On 8 April, the upper layer appeared to be less than 500 m deep, but was as much as 1900 m deep on 9 April. It is difficult to ascertain the variability in the nucleation mode size range (< 25 nm) for the elevated layers, as there may have been no particles, or too few for the instrument to obtain reliable counts.

Low instrument counts for the largest OPS sizes introduces sampling errors leading to an apparent large variability below concentrations of about 0.1 cm$^{-3}$ for all the layers.

### 3.1.3. Case I: Back-trajectory analysis

Figure 3 shows the origin of the air masses arriving at SMEAR II each day for Case I, based on HYSPLIT back-trajectories. The trajectories were separated into layers based on similarities in origin and the tracks in altitude over which they were advected. During 8 April most trajectories were from the north-west after spending some time over Greenland and the North Sea. Trajectories at the lowest altitudes, associated with the BL, came more from the Arctic Ocean (two lighter green shades in Figs. 3a-b). During 9 and 10 April, BL trajectories still arrived from the Arctic Ocean, but from further east.

The middle layer (yellow colour in Fig. 3) showed the largest variation in air mass origin, with trajectories arriving from Greenland on 8 April having been close to the surface two days before their arrival, and then from the Arctic Ocean on 9 and 10 April having descended from higher altitudes. The descent from higher altitudes implies drier air once it descends. This



change in the middle layer was seen in the RS profiles in Fig. 1a; the relatively moist middle layer on 8 April giving way to a relatively dry layer on 9 April.

The air mass origin for the upper layer (blue colour in Fig. 3) remained the same throughout Case I, arriving via Greenland having previously been over the North Atlantic. All the trajectories had previously been close to the surface, then elevated to altitudes above 3000 m for several days before reaching the station. This layer was always dominated by Aitken mode particles (see aerosol size distributions in Fig. 2); the aerosol particles did not have the right conditions to grow larger than the Aitken mode because the upper layer was relatively dry and had probably not been in contact with the ground-based emissions of aerosol precursor compounds for at least several days. Figures 3c and 3f also indicate that the interface layer (in red) on 10 April was at the altitude where trajectories arrived from two distinct spatial origins.

Overall, the air masses separated using backward trajectories corresponded very well with the layers recognized using the HSRL. The altitudes did not match perfectly, especially with regard to the thickness of the elevated layers and the height of the BL. During 9 and 10 April, for example, the trajectory BL height was lower than the BL seen from the HSRL and, consequently, the trajectory analysis suggested a thicker middle layer.

### 3.1.4. Case I: Evolution of the elevated layers

The evolution of the mean aerosol size distribution for the middle layer during Case I is shown in Fig. 4a. The aerosol particle concentrations remained similar across all the size ranges, except for sizes within the nucleation mode, and an increase in accumulation mode particle number concentrations was observed between 8 and 9 April. The change in accumulation mode is attributed to the change in air mass during this time. The lack of variation for most sizes indicates that there was essentially no mixing between the middle layer and the surrounding layers during this period. The large increase in nucleation mode particle concentrations for the afternoon flight of 9 April demonstrated that new particles could be formed in this layer.

In contrast, the upper layer exhibited a wide degree of variation in the aerosol size distributions across the three days of Case I (Fig. 4b). The distribution shape remained similar for particles larger than 300 nm, whereas the concentration varied, being the highest during 10 April and the lowest during the morning flight on 9 April. In the smaller particle range, the shape of the distribution was also similar, with an exception of 10 April, possibly due to the influence of an extra layer on April 10. This layer might have been mixing with air higher up in the troposphere.

### 3.1.5. Case I: Interface layer

We examined separately a thin 150 m boundary between the middle and upper layers during 10 April. This layer was characterized by strong scattering seen in the HSRL backscatter coefficient at around 1900 m (Fig. 2k) and it was located in



the area of RH change from 5 % to 15 %. Backward trajectories showed that this layer was a section where air masses of two different origin and height intersected. Panels of Fig. 5 demonstrates the total particle concentration measured by the CPC and uCPC during the ascent, the OPS data with a 10 s time resolution, as it was impossible to trace this interface layer only with the 2 minute time resolution data of the SMPS. The HSRL backscatter cross section data were averaged over the time of the

Cessna ascent. Peaks on all three panels occurred at the same height of 1900 m. On the first panel, showing the measurements of the CPC and uCPC, there was a peak in the total particle concentration, whereas concentrations of particles smaller than 10 nm, measured by the uCPC, did not change. A very high peak of 33 particles cm$^{-3}$ in comparison to the rest of the profile was seen on the total concentration of particles of 0.3 µm to 10 µm, measured by the OPS. When the size distribution was examined, it was found that the contribution to this peak came from particles in the diameter range of 300-500 nm. This thin layer could

be either a result of limited small-scale mixing between two layers, that were probably stable, or the result of large-scale transport of aged dust, especially since the low HSRL circular depolarisation values suggest more spherical particles. More data and further analyses are needed to understand the processes that lead to higher values of a backscatter cross section in these interface areas.

**3.2. Case II: cloudy during 22 August**

Figure 6 shows HSRL backscatter coefficient and circular depolarization ratio from 50 m to 4000 m for Case II. Backscatter values were, in general, higher than for Case I even though the aerosol number concentrations were similar for both cases, attributed to the much higher relative humidity in Case II resulting in significant aerosol hygroscopic growth. Fog was present from 6:00 to 9:30, severely attenuating the signal. As the fog lifted, the lidar was able to occasionally penetrate and detect the

deep residual layer above that extended to 1850 m in altitude. The residual layer showed low depolarisation and high backscatter values characteristic of a humid BL, in contrast to the layers above 1850 m. The BL started to mix into the residual layer during the morning at around 1100 and continued to deepen to at least 2000 m by mid-afternoon. Occasional cumulus clouds were formed from 1000 m in altitude, and were able to grow to at least 3000 m in altitude by late afternoon. Ground-based measurements from the SMEAR II station indicated that there was no NPF event at the surface during this case study.

Two flights were made with the Cessna during this day, one morning flight and one afternoon flight, and these are described in more detail below. A figure with similar panels as for Case I was generated for each flight.

**3.2.1. Case II: flight description**

The first Cessna flight during 22 August took place from 9:30 to 10:30. Even though the HSRL signal was often fully attenuated

by fog or low cloud, the height of the growing BL and the presence of the residual layer are clear in Fig. 7a-c. Both layers were also obvious in the radiosonde profile, launched one hour before the Cessna flight commenced. The BL RH was close to 100 %, hence the fog, with a constant specific humidity mixing ratio of about 7 g kg$^{-1}$. Between this layer and the residual



layer was an entrainment zone within which the specific humidity mixing ratio was decreasing rapidly with height. The residual layer above the entrainment zone also exhibited a relatively constant specific humidity mixing ratio of about 4.5 g kg$^{-1}$, from 700 m to 1850 m. Above 1850 m, the specific humidity mixing ratio decreased to about 1 g kg$^{-1}$; the profile indicated several additional layers, but these were difficult to distinguish in the HSRL backscatter coefficient and circular depolarisation ratio.

Together with the airborne in situ measurements, seven layers were diagnosed (Fig. 7c), classified as belonging to three main groups: first (green), middle (yellow) and upper (blue) layers. The first group comprised the BL, residual layer, and the entrainment zone. As a group, these layers displayed low concentrations of nucleation mode particles, but much higher concentrations in all other modes, relative to the layers above. The three layers in this group exhibited differences mostly in the Aitken mode: the BL and entrainment zone exhibited a peak at different sizes in the Aitken mode, whereas the residual layer exhibited a Hoppel minimum for sizes in the 80-100 nm region characteristic of cloud processing (Hoppel et al., 1990). The middle group was also separated into three layers, displaying differences in the nucleation and Aitken mode number concentrations. These layers corresponded well with the humidity structure seen in the radiosonde profile. The aerosol size distribution in the upper layer was similar to the one in the lowest layer in the middle group, except for lower concentrations below 30 nm.

The second Cessna flight on 22 August took place from 14:00 to 15:30 (Fig. 7d-f). Several cumulus clouds were present during the flight and can be seen in the HSRL backscatter coefficient (Fig. 7e). The BL had now grown to consume the residual layer from the previous day, but the radiosonde profile suggests that the BL was not as well mixed as one would expect for a classical BL – the profile of specific humidity mixing ratio was not constant and RH is not always increasing with height. Two tendencies are seen in the BL: a more mixed lower part up to about 1000 m where the cloud bases were, and a less mixed upper part. Three layers were identified in the airborne data: the BL and two middle layers. The shape of the aerosol size distribution in the BL was similar to the residual layer of the earlier flight, also displaying a Hoppel minimum (90 nm). The middle layers were separable with respect to Aitken mode particle concentrations, and could also be diagnosed from the radiosonde profiles. They corresponded with the middle layers seen during the descent of the morning flight. The upper layer of the morning flight was not detected during this flight, due to the limitations of the Cessna flight ceiling. Interestingly, nucleation mode particles were detected in the all layers during both flights of Case II.

### 3.2.2. Case II: Size distribution variability within layers

The elevated layers showed the same variability in the aerosol size distribution as was observed in Case I for particles smaller than 300 nm, but less variability in the accumulation and coarse modes. The BL exhibited more variability in the aerosol size distribution than was seen for the clear sky case (Case I), even during the afternoon flight (Fig. 7f). The radiosonde profiles show that the BL was not as well-mixed as in Case I, as the specific humidity mixing ratio was not constant with height (Fig. 7d), in strong contrast to the BL profiles seen in Fig. 3. The residual layer exhibited a profile of specific humidity mixing ratio



that was relatively constant (Fig. 7a), presumably a result of a well-mixed BL on the previous day, but this layer was no longer turbulent and also showed some variability in the size distribution. The BL was clearly convective, but the mixing was not homogeneous, indicated by the presence of deep cumulus clouds that were forming as a result of more organised updrafts.

### 3.2.3. Case II: Back-trajectory analysis

Figure 8 shows the origin of the air masses arriving at SMEAR II each day for Case II, based on HYSPLIT back-trajectories. As for Fig. 3, the trajectories were separated into layers based on similarities in origin and the altitude tracks over which they were advected. Similar to Case I, the air mass origins were from the North, but now travelled over the relatively warm Baltic sea before arriving at the station, where air in the BL could pick up moisture and become more humid. There was little change in altitude over the four days for the majority of the back-trajectories; however, by mid-afternoon, there were some trajectories inserted into the upper portion of the BL over Hyytiälä that had been close to the surface prior to their ascent over the Norwegian mountains between 06:00 and 18:00 EEST on 20 August (Fig. 8d; light green layer).

For this case, it would have been much harder to identify any layers based on back-trajectory analysis alone, since there was not much change in altitude over time, or in spatial origin. The set of trajectories that had been elevated from the surface (Fig. 8d; light green layer), would have been mixed into the BL by the time they reached the station.

### 3.2.4. Case II: Evolution of the layers

The BL aerosol size distribution measured during the afternoon flight resembled the distribution seen in the residual layer during the morning flight (Fig. 9a), with a Hoppel minimum suggesting cloud processing still visible for sizes around 70 nm. The afternoon BL displayed higher concentrations of nucleation mode particles than both the residual layer and morning BL, but decreased concentrations of particles above 500 nm. Both middle layers showed little change in the aerosol size distribution between the morning and afternoon flights, except for small differences in the nucleation mode for the first middle layer (Fig. 9b). This may be a result of occasional localised cloud-driven entrainment when cumulus clouds begun to extend into the first middle layer during the second flight. This was also indicated in the HSRL measurements (Fig. 6 and Fig. 7e) as the BL top was more diffuse (Fig. 7c) and quite variable even on short timescales, changing by as much as 1 km in 10 minutes. The presence of cumulus clouds suggests that the BL top was spatially heterogeneous.

### 4 Summary and conclusions

We present an analysis of aerosol layers over a relatively clean background measurement station based on a combined dataset comprising ground-based remote sensing observations, radiosonde profiles and airborne in situ measurements. The backscatter cross section coefficient and circular depolarisation provided by HSRL, together with the radiosonde humidity profiles, were





used for diagnosing layers and their evolution, while airborne SMPS and OPS measurements provided the aerosol size distribution from 10 nm to 10 μm within each of these layers at altitudes up to 3800 m.

Two case studies were chosen: a typical clear sky situation lasting for three consequent days, and a partly cloudy day. Several

elevated aerosol layers were detected in both cases. For the clear-sky case, the highest aerosol number concentrations were observed in the BL, for all modes. The radiosonde profiles indicated a classic well-mixed profile within the BL, which was also apparent in the low temporal variability in the measured aerosol size distributions. Outside the well-mixed BL, the temporal variability in the measured aerosol size distributions was usually much larger. The elevated aerosol layers showed size distributions with very similar shapes to the size distribution in the BL, but the typical number concentration in each layer

differed. The back trajectories suggested that most of the elevated layers had spent some time close to the surface previously, and that, since the air masses all had similar origin, and therefore aerosol source, the difference in number concentration was presumably due to the amount of dilution experienced through entrainment in each layer during transport.

Nucleation mode particles were observed in the elevated layers. Since the aerosol concentrations in one of the elevated layers

remained constant for several days with essentially no mixing observed, this suggests the potential for new particle formation occurring in the elevated layer at the same time as in the BL. In addition, a thin 'interface' layer was observed, between two distinct elevated layers, containing high concentrations of particles between 300 and 500 nm. Without chemical composition information, not available on these flights, it was not possible to determine whether this thin layer was a result of small-scale mixing between two adjacent layers, or whether these particles were the result of large-scale transport of smoke or dust,

especially since the low HSRL circular depolarisation values suggest more spherical particles.

In contrast to the clear sky case, the BL for the cloudy case did not appear to be as well mixed, even though a convective BL, expected to promote mixing, was clearly present. This was evident in both the radiosonde profile, and in the larger variability exhibited in the aerosol size distributions measured in the BL, implying that the organised convective structures present were

responsible for the heterogeneity seen in the BL. Evidence for cloud processing of aerosol particles was also seen in the BL but the amount of processing varied, presumably again due to the specific nature of the updrafts and downdrafts resulting in BL mixing that was not homogeneous. Conversely, some of the convective plumes reached sufficient altitudes in the afternoon to provide a degree of mixing in the lower elevated layer, but this mixing was not spatially homogeneous.

We computed back trajectories every 50 m in altitude from HYSPLIT to assess whether the vertical layer structure could be explained in terms of air mass origins, determine whether individual elevated layers could be identified from the back trajectories, and test how reliable the arrival heights of the back trajectories were for each layer. The results show that layers diagnosed from HSRL could be identified from back trajectories, with the aid of RS humidity profiles, although the arrival heights did not always coincide. Not all layers could be identified from the back trajectories alone. The conclusion is that a



combination of RS, HSRL and back trajectories gives much more confidence in determining the air mass origin and vertical layer extent when interpreting local measurements in terms of long-range transport.

**Acknowledgements**

5    This work was supported by the Academy of Finland Centre of Excellence program (project no 307331). HSRL and radiosonde data were obtained from the Atmospheric Radiation Measurement (ARM) Climate Research Facility, a U.S. Department of Energy Office of Science user facility sponsored by the Office of Biological and Environmental Research. We are very grateful for the collaboration with ARM during the BAECC campaign in Hyytiälä, Finland. We also acknowledge the NOAA Air Resources Laboratory (ARL) for the provision of the HYSPLIT transport and dispersion model used in this publication.

Atmospheric Radiation Measurement (ARM) Climate Research Facility. 2014, updated hourly. High Spectral Resolution Lidar (HSRL). 2014-04-08 to 2014-08-22, ARM Mobile Facility (TMP) U. of Helsinki Research Station (SMEAR II), Hyytiala, Finland; AMF2 (M1). Compiled by B. Ermold, E. Eloranta, J. Garcia and J. Goldsmith. Atmospheric Radiation Measurement (ARM) Climate Research Facility Data Archive: Oak Ridge, Tennessee, USA. Data set accessed 2014-09-25 at
http://dx.doi.org/10.5439/1025200.

Atmospheric Radiation Measurement (ARM) Climate Research Facility. 2014, updated hourly. Balloon-Borne Sounding System (SONDEWNPN). 2014-04-08 to 2014-08-22, ARM Mobile Facility (TMP) U. of Helsinki Research Station (SMEAR II), Hyytiala, Finland; AMF2 (M1). Compiled by D. Holdridge, J. Kyrouac and R. Coulter. Atmospheric Radiation Measurement (ARM) Climate Research Facility Data Archive: Oak Ridge, Tennessee, USA. Data set accessed 2014-09-25 at
http://dx.doi.org/10.5439/1021460.

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





Table 1. Case study I: flight times, diagnosed layer parameters (in m) and new particle formation start time

| Date | Time EEST | BL height | MidL height | MidL depth | MidLII | UppL height | UppL depth | NPF start time in Hyytiälä |
|---|---|---|---|---|---|---|---|---|
| 8 April | 16:00-17:30 | 1600 | 1600-2400 | 800 | 2400-3100 700 | 3100-3350 | 250 | 10:00 |
| 9 April 1 | 11:00-12:30 | 1000/1200 | 1000-1500 | 500 | | 1500-3400 | 1900 | 11:30 |
| 9 April 2 | 16:00-17:40 | 1400 | 1400-1800 | 400 | | 1800-3400 | 1600 | 11:30 |
| 10 April | 13:45-15:30 | 1200 | 1200-2000 | 800 | | 2000-2800 | 800 | 09:30 |



**Figures**

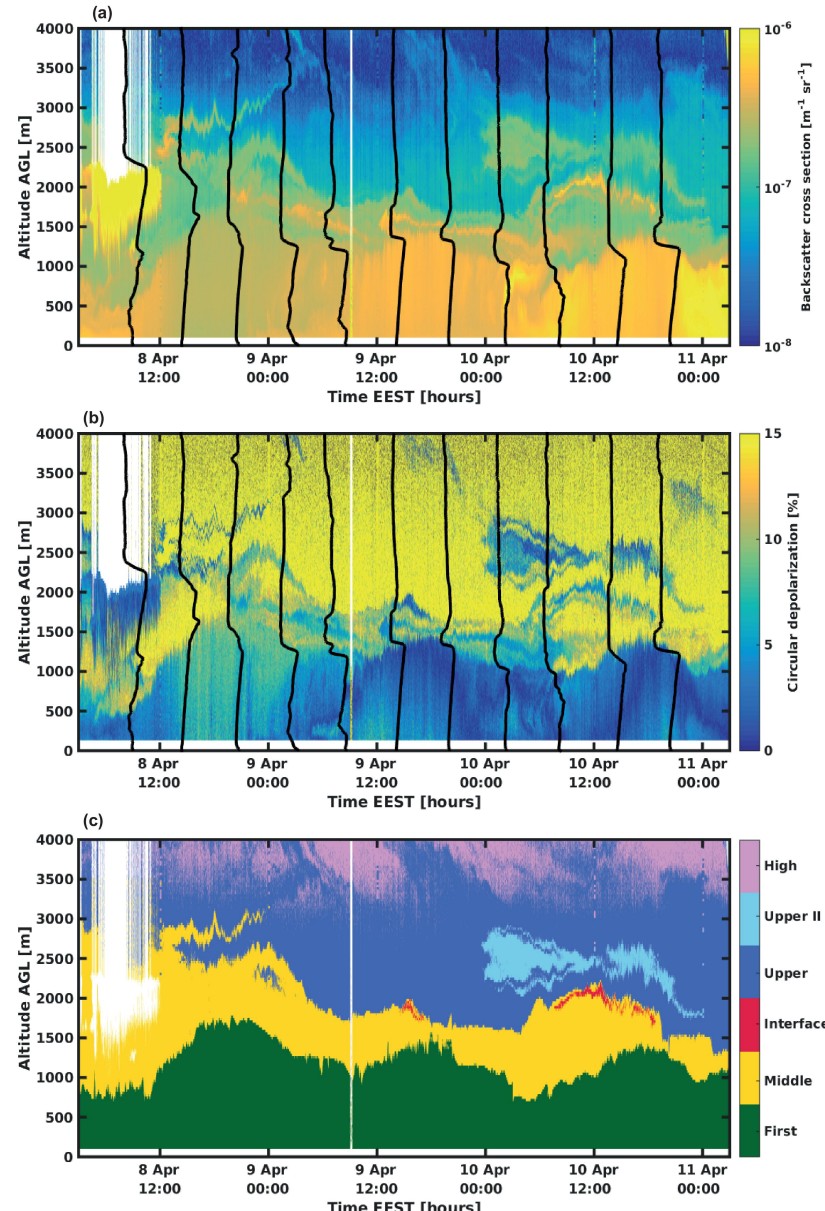

**Figure 1: Case I: a) HSRL backscatter coefficient and b) HSRL circular depolarization over Hyytiälä, Finland during 8-10 April 2014, with 6-hourly radiosonde relative humidity profiles superimposed in black. Larger values of the backscatter cross section indicate higher aerosol number concentration or larger particles; larger depolarisation values suggest less spherical particles. c) Layers diagnosed from gradients in backscatter or depolarization, with the First layer (green) comprising both boundary layer and residual layer. White pixels indicate no valid measurement due to the presence of cloud and subsequent attenuation (before midday on 12 April), or due to calibration period (around 10 EEST on 9 April).**



**Figure 2: Four panels for each flight during Case I: a-c) 8 April, d-f) morning flight on 9 April, g-i) afternoon flight on 9 April, j-l) 10 April. Left panels show relative and specific humidity profiles measured by the radiosonde launched closest in time to the flight. Centre panels display HSRL backscatter coefficient, with Cessna flight altitude superimposed in black. Right panels present layer-averaged aerosol size distributions from combined SMPS and OPS measurements for diagnosed layers. Mean and one standard deviation are shown for each layer. Grey lines show limits of the aerosol modes: nucleation, Aitken, accumulation and coarse.**





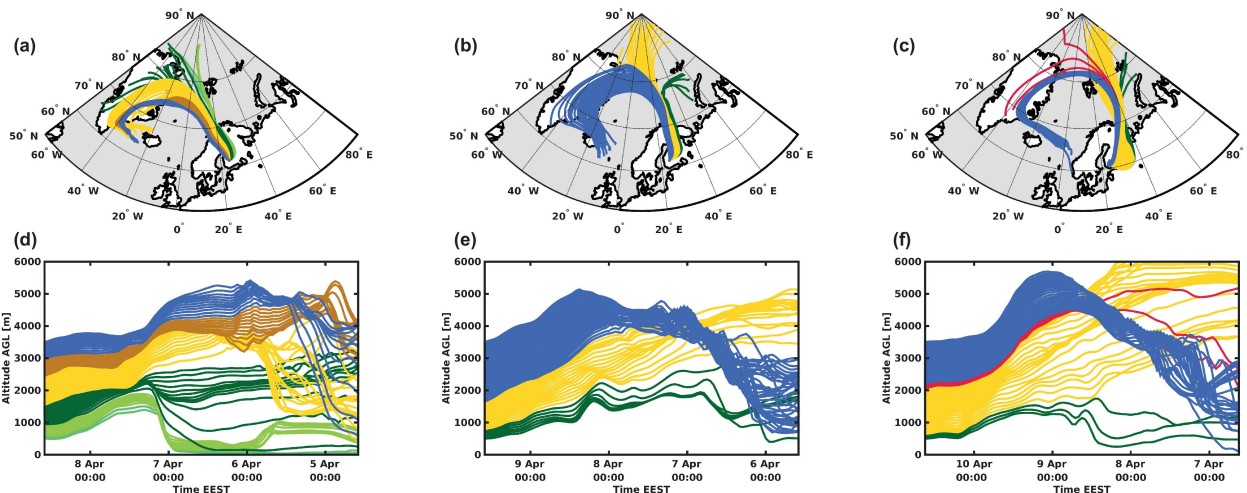

**Figure 3: HYSPLIT 96-hour backward trajectories arriving at Hyytiälä calculated every 50 m in altitude from 500 m to 3500 m. Panels a-c show spatial coverage and panels d-f display the trajectory altitude over time. Trajectories with similar origin/altitude properties are combined into layers with the same colours as the layers identified with HSRL in Fig. 1c. Panels a), d) show trajectories arriving on 8 April at 17:00, panels b), e) on 9 April at 17:00 and panels c), f) on 10 April at 14.00.**

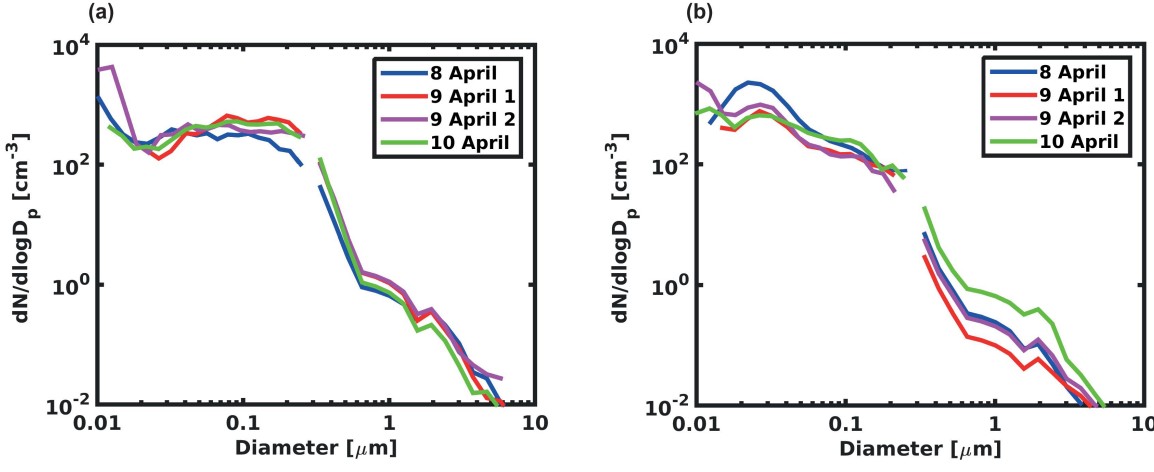

**Figure 4: Evolution of the mean aerosol size distribution in Case study I for a) middle and b) upper layers. Minimal change in the size distribution during three consequent days implies that the middle layer did not mix with surrounding air. Upper layer does show changes over time, suggesting some entrainment.**





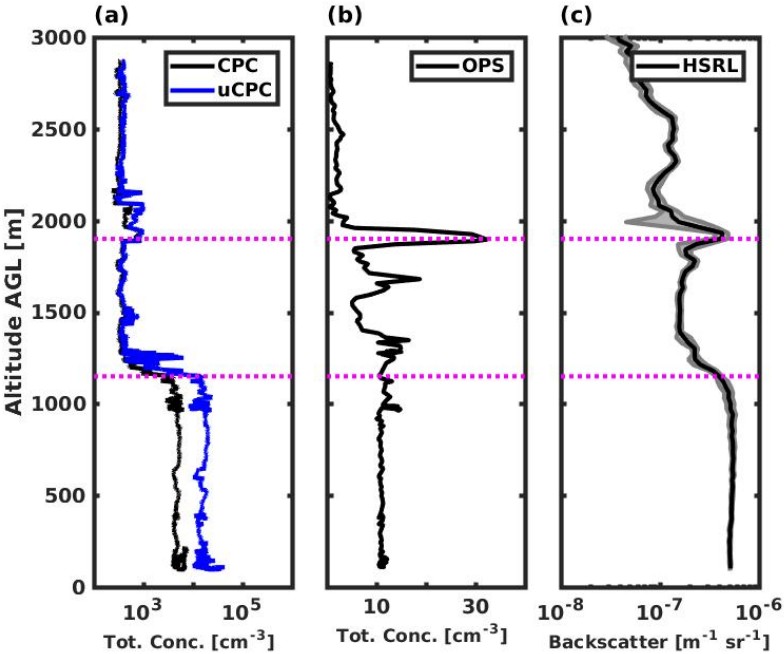

**Figure 5: Vertical profiles of a) total particle concentration measured by uCPC and CPC, b) total particle concentration measured by OPS, c) HSRL backscatter coefficient averaged over the time taken for the Cessna ascent, over Hyytiälä, Finland, on 10 April 2014. All panels show a presence of a thin layer at 1900 m (dashed magenta line) with OPS indicating enhanced contribution of larger (0.3 -10 μm) particles.**





**Figure 6: Case II: a) HSRL backscatter coefficient and b) HSRL circular depolarization over Hyytiälä, Finland during 22 August 2014, with 6-hourly radiosonde relative humidity profiles, and the two Cessna flight altitude tracks superimposed in black. White pixels indicate no valid measurement due to the presence of cloud and subsequent attenuation above or maintenance periods.**



**Figure 7: Same as Fig. 2 except for Case II with a-c) morning flight d-f) afternoon flight.**





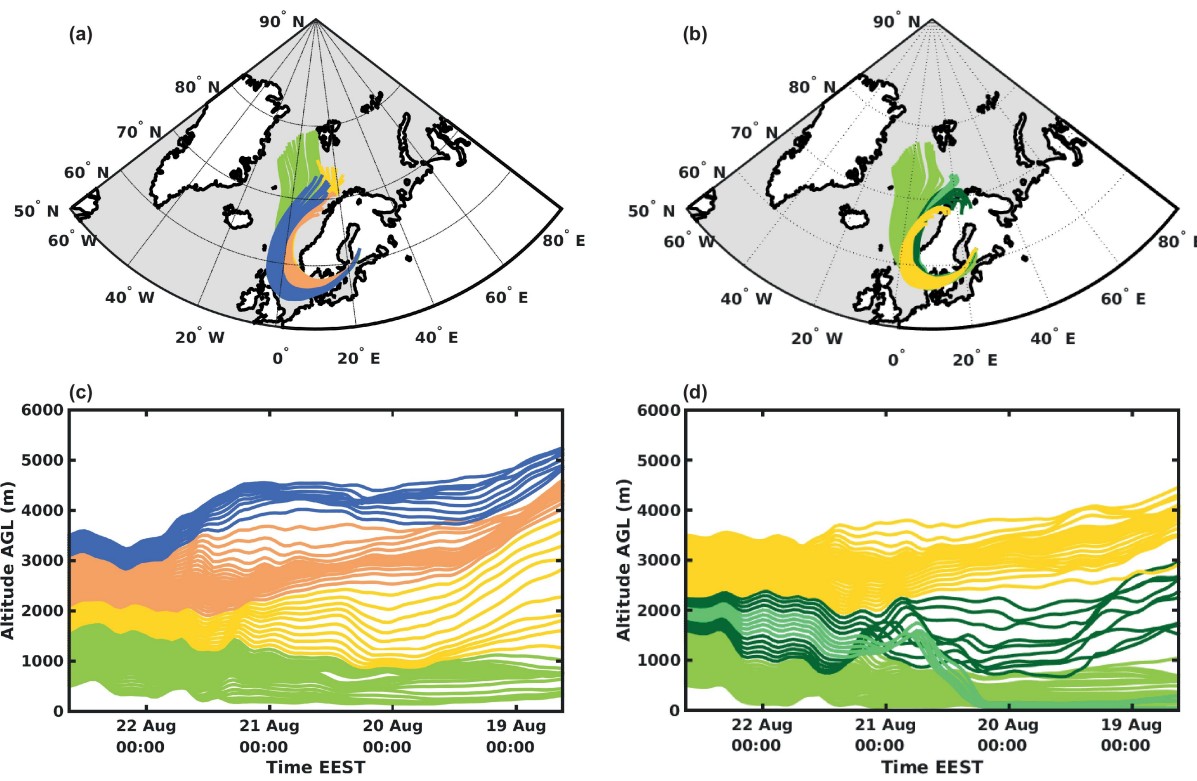

**Figure 8: Same as Fig. 3 except for Case II with a), c) show trajectories arriving at 10:00 and panels b), d) at 14:00.**





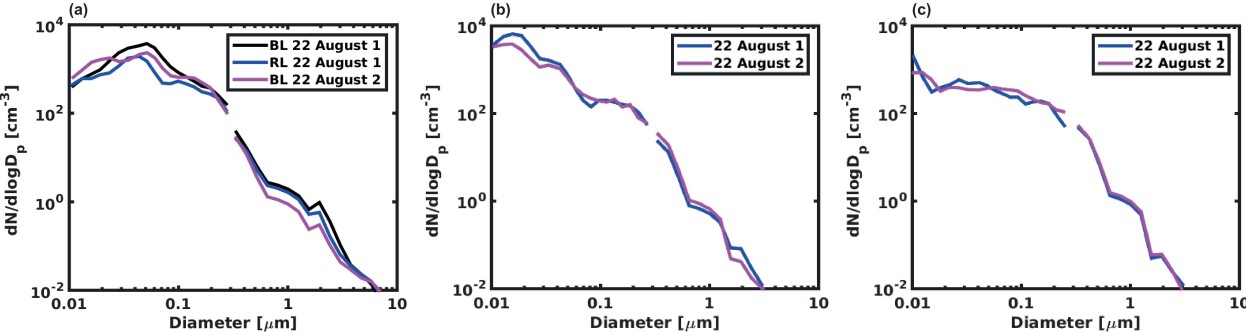

**Figure 9: Evolution of the mean aerosol size distribution in Case II for a) boundary layer b) first middle layer and c) second middle layer.**