# Peer review of "Combining airborne in situ and ground-based lidar measurements for attribution of aerosol layers"

_Atmospheric Chemistry and Physics, 2017_

## Referee Comment (RC1) · Anonymous Referee #3 · 27 Jan 2018

I have reviewed "Combining airborne in situ ground based lidar measurements for attribution of aerosol layers" by Nikandrova et al. The work presents a combination of results derived from a range of instrument systems deployed during the BAECC (Biogenic Aerosols—Effects on Cloud and Climate) campaign conducted at a field site in southern Finland. The work presents evidence of lofted layers of aerosol and examines differences in the aerosol size distributions, lidar backscatter, and lidar depolarization ratio for two case studies. Overall, the results presented in the manuscript should be interest to the larger research community. I believe that he manuscript will be acceptable for publication after addressing a few significant concerns: 1. The work makes a point of using relative humidity (RH) to help define layers and to better understand some of

the differences in the observed size distributions. The authors point out that the ambient RH was measured near the aerosol inlet, but there is no discussion of how the RH might change as the particles move through the system. Perhaps the impact is small, but it should be addressed in the manuscript in some fashion. In addition, there is little detail given about the RH measurements themselves (for example, what instrument is used to make the measurements). 2. The authors describe that the atmospheric thermodynamic structure is hard to interpret for Case II. Based on the figure derived from the radiosonde, it looks like it could be a more typical profile with clear subcloud layer to an altitude of 750 m, and then a cloud layer from 750 m to approximately 1700 m. It could be helpful to examine the potential temperature profile in addition to the humidity in this case.

Minor comments: 1. Page 1, line 14: The acronym DOE (rather than DoE) is generally used for the Department of Energy 2. Page 1, line 17 (and other locations): The terms "low" and "high" are used throughout the manuscript when referring to variability, aerosol concentration, and other meteorological variables. It is better usage to use "small" and "large" unless one is referring to differences related to altitude. 3. Page 2, line 15. The authors might want to consider adding a reference to Wang et al. (2016), a Nature paper also looking at vertical transport of aerosol. 4. Page 3, lines 2-6. Muller et al. (2014, AMT) also showed comparisons of aerosol microphysical properties derived from HSRL with in situ data. 5. Page 2, line 20. Suggest using "aerosol" or "particles" rather than "aerosol particles". 6. Page 5, line 16. Does the reference to BLs really mean convective BLs or all BLs? 7. Page 5, lin3 24. The manuscript cites the work of Laakso et al. (2004). Are their results for the same geographic area? 8. Page 6 (figure 1). The humidity profiles in the figure are hard to interpret due to the lack of a scale. Is there a way to add a scale or axes that would not make the figure too hard to read? 9. Page 6. line 21. It is not clear to me how you determined the interface zone, and what you really mean by the term. It is addressed a bit later in the manuscript, but some additional explanation here would be helpful when the term is introduced. 10. Page 7 (and Figure 2 and 7). Is there a way to mark the specific layers on the plots of aerosol

backscatter? I know that there is a table, but it could be helpful to add the information to the figure (assuming that the figure remains legible). 11. Page 8, lines 4-5. Should a reference or references for NPF be added? 12. Page 8, lines 20-21. I can see how this sentence is needed, but it seems to be just tacked onto the end of the section. 13. Page 10, lines 9-10. How would small scale mixing lead to the behavior that is shown in the figure? 14. Page 11, line 10. Suggest using the same units as shown on the figure. 15. Page 12, line 2. The text mentions deep convection, but can that be safely said from the data that has been presented so far? Wouldn't the HSRL have issues seeing the cloud-top height?

---

## Referee Comment (RC2) · Anonymous Referee #4 · 12 Feb 2018

Review of : Combining airbone in-situ and ground based Lidar measurements for attribution for attribution of aerosol layers.

By Nikandrova et al.

The authors describe two case studies (clear sky and cloudy sky) observed over the SMEAR-II station during a field campaign in 2014. The authors used airborne measurements (mostly in-situ size distributions) associated to ground based HSRL Lidar. This manuscript is of interest for the scientific community but need major revisions before submission to ACPD.

Fist of all, the aim of the paper is pretty vague: ' investigate aerosol layers in a rural environment' and need to be clarified. This paper is showing size distribution differences that occur within each layer of the atmosphere as a function of time. The authors interpret each increase of the fine particle number concentration as a nucleation event within each layer. However, the differences of Aitken, Accumulation and Coarse number concentrations are only pointed out.

The conclusions of this paper needs some work : The comparison of the RH and HSRL profiles with HYSPLITT results are most of the times in good agreement but the ''heights did not always coincide''. These height differences are not expressed in the main part of the paper and should probably be... A brief presentation of the Hysplitt model and especially the resolution of the data input of the model could help the authors to interpret these differences. Also the last conclusion of the paper is that the synergy between radiosounding, LIDAR and back trajectories gives more confidence in determining the air mass origin. Is this really the main conclusion ?

Last, the authors state: 'Evidence for cloud processing of aerosol particles was also seen in the BL but the amount of processing varied [...]'. The authors are showing Hoppel minimum that could be related to cloud processing but it's not supported by real evidence. It could also be due to different sources of aerosol with one source quite close to the instrumental site ? Moreover, I believe you can't talk about the 'amount of processing'...

Minor corrections:

P3 L 4 : Not well said. Please rephrase

Page 3 L15 : Needs to add references to support that like Crumeyrolle et al., 2010; Rose et al., 2015a, Berland et al., 2016.

P5 L25 : Please add explanations. I don't want to read Laakso et al. To understand what you did. The GF is usually dependant of the different compounds present in the aerosol. So how did you get this information ?

[Figure]

Figure 2 : You are always refering to the mode you define P5. Could you add on your size distribution plots the limits of each mode (nucleation, Aitken, Accumulation, Coarse). It would help the reader. No error bars on the Figure 2i within the small particles range for the middle layer ?

P7 L25 : Hard to tell cause there are no measurements of the fine particle number concentration within the middle layer...

Section 3.1.2 : If you are talking about erors you need to state the number of SD you used to get the average showed in figure 2. ..

P9 L27-29 : Please tell us more about the difference you see cause it's not obvious for me.

P10 L7 : 'A very high peak' : could you add in comparison to the rest of the profile ?

P10 L10 : Smoke or Dust are not known to be spherical particles ...

P11 L14 : below 100nm instead of 30nm

P11 L 20 : If you are implying that the cloud base is playing a role in the mixing efficiency be more clearer

P11 L 22 : Please add 100nm to show the reader where the Hoppel minimium is located.

P12 L17 : around 100nm replace with around 70nm

P12 L 19 : Any interpretation why there is less particles above 500nm ?

P12 L22 : Do you mean that nucleation occurs over the cloud top ? Please add references to support this.

Figure 7. Not able to distinguish the 3 green shades...

Figure 9 : No error bars : Does it mean that you used only one spectra. If yes than it needs to be stated somewhere.

[Figure]

---

## Referee Comment (RC3) · Anonymous Referee #2 · 12 Feb 2018

The manuscript "Combining airborne in situ and ground based lidar measurements for attribution of aerosol layers" focuses on investigating different layers present in the troposphere up to 3500 m. For this purpose, they combine aerosol particle size distribution data recorded on board of a research airplane with ground-based High Spectral Resolution Lidar (HSRL), radiosonde profiles and air-mass back trajectory analysis within the BAECC campaign which took place in Southern Finland 2014. The data is presented for two main case studies recorded at the same location but with differing meteorological conditions. The presence of several lofted layers was seen and compared to findings from the back trajectory analysis.

[Figure]

I recommend the paper for publication in ACP after the following comments have been addressed:

P4, Chapter 2.1- HSRL: what is the minimum altitude that can be measured?

P4, line 32: What type of RH sensor was employed and what is the expected uncertainty?

P5, Chapter 2.2: It is stated that the SMPS data is corrected for elevated RH in the ambient with a certain GF. Is this correction implemented as a function of height, meaning that for each altitude the actual RH that was measured was used to determine the GF? What about the influence of elevated RH on the optical properties? It is not stated in the paper which index of refraction was used to determine the optically measured size distribution! As a correction for the SMPS is introduced I would strongly suggest to also apply a correction to changes in the index of refraction of the particles and adjust the actual size range measured by the OPS.

Figures 2&7: Could you add lines for the different layers in the plots depicting the back-scatter cross sections? It is not very clear where the boundaries were chosen.

P7, line 9: Where can the mentioned depolarization be seen?

P8, Chapter 3.1.3: Which data was used for the HYSPLIT trajectories (GDAS etc.?) and what resolution was employed? These two things can have a strong influence on the analysis.

P8, line 30: "close to surface" – I am a bit confused by this statement as from Fig. 5d the lowest height visible is around 1000 m, and I would not refer to that as "close to surface". Could you rather state the actual altitude range? Such phrasing is also used later, and I would suggest changing that as well (for blue lines).

P9, lines 12-15: Can some possible reasons for the not-matching altitudes between HYSPLIT and the measurements be pointed out?

P9, line 26+27: What is meant here by "the smaller size range"? I am also confused by the change mentioned for 10th of April. What is it referred to? I cannot see a clear difference between the lines in Fig. 4b?

Specific comments:

P2, line 3: change "vary" to "varies"

P4, line 23: change "stainless tube" to "stainless steel tube"

P5, line 4: change "campaigns" to "campaign"

P5, line 20: add "the" before "Cessna"

P10, line 2: change to "air masses of two different origins and heights intersected. The panels of Fig. 5 demonstrate the. . . "

---

## Referee Comment (RC4) · Anonymous Referee #1 · 20 Feb 2018

This is a nice workup of case studies using multiple sources of data (lidar profile measurements, relative humidity from radiosondes, in situ size distributions, and back-trajectory analysis). Although it is somewhat limited in scope, I think the analysis successfully uses these multiple disparate data sources to gain a deeper understanding of the atmospheric layers in the case studies. The figures are informative and well constructed for showing correspondence between different measurement types and for illustrating interesting aspects of the case studies. I recommend publication after addressing a few points.

Specific comments:

[Figure]

Page 2, line 30. Delete "at higher latitudes". Smoke aerosol is not limited to high latitudes.

Page 4, line 13. "the cross-polarization channel measures the degree of circular polarization". I think this should probably be reworded. I don't think just one channel by itself can measure the degree of polarization; it must be compared to another channel. A related question: what is the polarization state of the combined channel? That is, does the polarization split occur before or after the Rayleigh-Mie split?

Page 4, line 14. I would have liked to look up the answer to my previous question in the quoted reference (Goldsmith 2016) but it isn't in the bibliography.

Page 4, line 24. What is the particle size cutoff of the inlet?

Page 5, line 29-31. Are these quoted sizes radius or diameter?

Page 10, line 11. "aged dust, especially since the low HSRL circular depolarization values suggest more spherical particles". I am confused by this sentence. Dust, even aged dust, would be expected to be dominated by non-spherical particles. Either I'm misunderstanding the intent of the sentence (in which case, please reword) or else you are suggesting that aged dust would be expected to have spherical depolarization values similar to what's observed. If that's the intent, please include more discussion and references to support this idea.

Figures 1 seems to show enhanced depolarization during the time period selected for the case study (8 April). Any comment about what this might indicate?

Lidar ratio can give important insight into aerosol type and therefore would potentially provide another useful clue for analyzing the case studies. Also, there is significant interest in the aerosol lidar community in cataloging lidar ratio for different aerosol scenarios. HSRL measures backscatter and extinction separately and therefore includes lidar ratio. Why not include lidar ratio in Figures 1 and 6 and in the analysis?

Page 10, line 23 discusses the depth of cumulus clouds. Since these block the laser

light, it's not clear how you estimate the top-heights of these clouds. Please explain.

In the discussion section, please include more discussion of the proposed mechanisms for new particle formation in the particular cases discussed. I realize there are no measurements available to explain this definitively, but I think some more specific discussion of possibilities supported by literature references would be helpful. Specifically, you discuss new particle formation in the boundary layer for case 1 and then use back-trajectory analysis to infer that the airmass originated over the Arctic Ocean. Does this mean that the new particle formation occurred over the Arctic Ocean? Was this area covered by sea ice? You also suggest that new particle formation occurred in the elevated layer at the same time. What are published mechanisms for new particle production over sea ice and in elevated layers that would be consistent with these observations?

Typos, etc.

Page 4, line 14. "Goldsmith" misspelled

Page 4, line 24. Is this liters per minute? Can the "L" be capitalized? It looks like a "one".

Page 5, line 14. "for the algorithm" is not clear. Do you mean for the layer-detection algorithm?

Page 5, line 18. "most often indicate edges of layers". Fragmented sentence.

Page 7, line 1. "this layer" is not clear, since you mention four layers. Which layer?

Table 1. Please explain acronyms in the table caption (particularly "NPF"). Also, the formatting of the "MidLII" column is strange in that it is unlike any other column in having both the height and depth. I realize this is to save space since there is only one layer. Another possibility that might be clearer is removing the "MidLII" column and putting two sets of measurements (separated by a comma) in that row of the "MidL height" and "MidL depth" columns.

Figures 2 and 7, the annotations are hard to read. Repeating the information from the color legend in the caption would help. It would also be useful to indicate the layer boundaries as lines or markers on the humidity profile or lidar curtain so that it would be more immediately obvious where the in situ size distributions are applicable.

Also, it would be useful to make the axis labels bigger in Figures 2, 3, 7, 8 and 9.

There seems to be a rendering or smoothing artifact in the lidar curtain in Figure 2e that shows as a series of horizontal lines where the lidar backscatter profile does not change for 15 or 20 minutes between 11:50 and 12:10.

---

## Author Comment (AC1) · 3 May 2018

The manuscript "Combining airborne in situ and ground based lidar measurements for attribution of aerosol layers" focuses on investigating different layers present in the troposphere up to 3500 m. For this purpose, they combine aerosol particle size distribution data recorded on board of a research airplane with ground-based High Spectral Resolution Lidar (HSRL), radiosonde profiles and air-mass back trajectory analysis within the BAECC campaign which took place in Southern Finland 2014. The data is presented for two main case studies recorded at the same location but with differing meteorological conditions. The presence of several lofted layers was seen and compared to findings from the back trajectory analysis.

I recommend the paper for publication in ACP after the following comments have been addressed:

**Response to comments from Anonymous Referee #2**

We thank the referee for the constructive comments to help us to improve the manuscript. Below please find our answers to the comments.

Comments:
P4, Chapter 2.1- HSRL: what is the minimum altitude that can be measured?
Added to the text: 'The HSRL instrument provides profiles from around 50 m up to 30 km in altitude.'

Full overlap between the transmitted laser beam and the telescope is reached at around 3 km, however values below this altitude can be used qualitatively down to the minimum altitude with their uncertainty increasing as the overlap decreases. Note that the retrieved backscatter coefficient and lidar depolarization ratio values are usually less affected by the overlap issue as they are derived by taking ratios of two channels (ie combined and molecular channels, combined and cross polarization channels), assuming all channels follow similar paths in the detection chain.

P4, line 32: What type of RH sensor was employed and what is the expected uncertainty?
Added to the text: a relative humidity sensor (Rotronic HygroClip-S, accuracy 0.8 % at 23 °C)

P5, Chapter 2.2: It is stated that the SMPS data is corrected for elevated RH in the ambient with a certain GF. Is this correction implemented as a function of height, meaning that for each altitude the actual RH that was measured was used to determine the GF?
-Yes. Added to text: The correction for GF was implemented as a function of height for the SMPS data.
What about the influence of elevated RH on the optical properties? It is not stated in the paper which index of refraction was used to determine the optically measured size distribution! As a correction for the SMPS is introduced I would strongly suggest to also apply a correction to changes in the index of refraction of the particles and adjust the actual size range measured by the OPS.
-Index of refraction for water droplets (1.334) was used for OPS measurements.
During Case Study I, the relative humidity values were lower than 50 %, so that any changes in refractive index are expected to be minor. During Case Study II, it is true that RH values were much higher at some altitudes, and that this may induce some changes in the refractive index that is assumed constant for the OPS data. However, we have no chemical composition measurements from the aircraft from which the real refractive index could be determined, hence we may introduce more errors by varying the refractive index. In this study we are more concerned with identifying that there are significant differences in the aerosol size distribution from layer to layer, for which we believe our simple assumption of a single refractive index is still appropriate.

Figures 2&7: Could you add lines for the different layers in the plots depicting the back-scatter cross sections? It is not very clear where the boundaries were chosen.
- Lines were added to the left most panels of figures 2 and 7.
The text was modified: The ambient size distributions were then grouped by similarities in the size distribution and taking into account the layer boundaries found in the HSRL data.

P7, line 9: Where can the mentioned depolarization be seen?
Added to the text: Fig 1b

P8, Chapter 3.1.3: Which data was used for the HYSPLIT trajectories (GDAS etc.?) and what resolution was employed? These two things can have a strong influence on the analysis.
- Added to the text: The National Center for Environmental Prediction (NCEP) Global Data Assimilation System (GDAS) dataset with 1 degree resolution was used for the meteorological input to the model.

P8, line 30: "close to surface" – I am a bit confused by this statement as from Fig. 5d the lowest height visible is around 1000 m, and I would not refer to that as "close to surface". Could you rather state the actual altitude range? Such phrasing is also used later, and I would suggest changing that as well (for blue lines).
-Modified as suggested

P9, lines 12-15: Can some possible reasons for the not-matching altitudes between HYSPLIT and the measurements be pointed out?
- Added to the text: Errors in trajectories (particularly in the vertical) arise from the difficulties that the meteorological models providing the wind fields have in accurately representing vertical motion, turbulence and other sub-grid scale features (Stohl et al., 2001, Riddle et al., 2006, Hoffmann et al., 2016).

Riddle, E. E., P. B. Voss, A. Stohl, D. Holcomb, D. Maczka, K. Washburn, and R. W. Talbot (2006), Trajectory model validation using newly developed altitude☐controlled balloons during the International Consortium for Atmospheric Research on Transport and Transformations 2004 campaign, *J. Geophys. Res.*, 111, D23S57, doi:10.1029/2006JD007456.

Stohl, A., L. Haimberger, M. P. Scheele, and H. Wernli. "An intercomparison of results from three trajectory models." *Meteorological Applications* 8, no. 2 (2001): 127-135.

L. Hoffmann, T. Rößler, S. Griessbach, Y. Heng and O. Stein, Lagrangian transport simulations of volcanic sulfur dioxide emissions: Impact of meteorological data products, *Journal of Geophysical Research: Atmospheres*, **121**, 9, (4651-4673), (2016).

P9, line 26+27: What is meant here by "the smaller size range"? I am also confused by the change mentioned for 10th of April. What is it referred to? I cannot see a clear difference between the lines in Fig. 4B?
-This paragraph is rephrased to be clearer: For particles smaller than 300 nm, the shape of the size distribution and the number concentrations changed from day to day.
For particles larger than 300 nm, while the number concentration varied, the shape of the distribution remained similar across all 3 days.

Specific comments: - changed as suggested
P2, line 3: change "vary" to "varies"
P4, line 23: change "stainless tube" to "stainless steel tube"
P5, line 4: change "campaigns" to "campaign"
P5, line 20: add "the" before "Cessna"
P10, line 2: change to "air masses of two different origins and heights intersected. The panels of Fig. 5 demonstrate the
...

---

## Author Comment (AC2) · 3 May 2018

I have reviewed "Combining airborne in situ ground based lidar measurements for attribution of aerosol layers" by Nikandrova et al. The work presents a combination of results derived from a range of instrument systems deployed during the BAECC (Biogenic Aerosols Effects on Cloud and Climate) campaign conducted at a field site in southern Finland. The work presents evidence of lofted layers of aerosol and examines differences in the aerosol size distributions, lidar backscatter, and lidar depolarization ratio for two case studies. Overall, the results presented in the manuscript should be interest to the larger research community.

I believe that he manuscript will be acceptable for publication after addressing a few significant concerns.

**Response to comments from Anonymous Referee #3**

We thank the referee for the constructive comments to help us to improve the manuscript. Below please find our answers to the comments.

1. The work makes a point of using relative humidity (RH) to help define layers and to better understand some of the differences in the observed size distributions. The authors point out that the ambient RH was measured near the aerosol inlet, but there is no discussion of how the RH might change as the particles move through the system. Perhaps the impact is small, but it should be addressed in the manuscript in some fashion. In addition, there is little detail given about the RH measurements themselves (for example, what instrument is used to make the measurements).

Added to the text: a relative humidity sensor (Rotronic HygroClip-S, accuracy 0.8 % at 23 °C)

For the SMPS measurements the change in the RH in the system was not an issue, as the air was dried before aerosol size distribution was measured (already mentioned in the text on p. 5). As for OPS measurements, RH higher than 40% could accelerate hygroscopic growth. Particles spend some tens of seconds in the sampling line. Humidity inside the cabin was lower than outside (higher temperature inside) so hygroscopic growth should have not taken place. Most of our case studies have lower than 50 % ambient humidity.

2. The authors describe that the atmospheric thermodynamic structure is hard to interpret for Case II. Based on the figure derived from the radiosonde, it looks like it could be a more typical profile with clear subcloud layer to an altitude of 750 m, and then a cloud layer from 750 m to approximately 1700 m. It could be helpful to examine the potential temperature profile in addition to the humidity in this case.

We have now examined the potential temperature, and we still think that the thermodynamic structure during this day was not easy to interpret. As seen from the profile, there are no clear sublayer boundaries:

[Figure]

Minor comments:

1. Page 1, line 14: The acronym DOE (rather than DoE) is generally
used for the Department of Energy
-Changed as suggested to DOE

2. Page 1, line 17 (and other locations): The terms "low" and "high" are used throughout the manuscript when referring to variability, aerosol concentration, and other meteorological variables. It is better usage to use "small" and "large" unless one is referring to differences related to altitude.
-Changed as suggested from 'low/high' to 'small/large' for the term variability (page 1, 8, 12). We think 'low/high' is appropriate to use for aerosol concentration.

3. Page 2,line 15. The authors might want to consider adding a reference to Wang et al. (2016), a Nature paper also looking at vertical transport of aerosol.
-Added suggested reference

4. Page 3, lines 2-6. Muller etal. (2014, AMT) also showed comparisons of aerosol microphysical properties derived from HSRL with in situ data.
-Added suggested reference

5. Page 2, line 20. Suggest using "aerosol" or "particles" rather than "aerosol particles".
-We think "aerosol particles" is the best word to use when discussing aerosol particles in this case.

6. Page 5, line 16. Does the reference to BLs really mean convective BLs or all Bls?
- The reference is to all BLs, and on a clear day the BL is usually convective.

7. Page 5, lin3 24. The manuscript cites the work of Laakso et al. (2004). Are their results for the same geographic area?
-Yes, they are. Added to the text: a growth factor (GF) calculated for a boreal forest environment using measurements from SMEAR II station.

8. Page 6 (figure 1). The humidity profiles in the figure are hard to interpret due to the lack of a scale. Is there a way to add a scale or axes that would not make the figure too hard to read?
-Figure 1 was intended to show the general overview of the situation during three days, while humidity profiles for case studies could be seen on Figure 2 with scales.

9. Page 6. line 21. It is not clear to me how you determined the interface zone, and what you really mean by the term. It is addressed a bit later in the manuscript, but some additional explanation here would be helpful when the term is introduced.
-Added to the text: The interface zone was a shallow zone situated at the boundary between two more substantial layers and was characterized by large backscatter values and depolarization values different from the surroundings. No corresponding thin layer was detected in the humidity profiles, whether from the radiosonde or aircraft.

10. Page 7 (and Figure 2 and 7). Is there a way to mark the specific layers on the plots of aerosol backscatter? I know that there is a table, but it could be helpful to add the information to the figure (assuming that the figure remains legible).
-Specific layer boundaries were added to the RS profiles panels of Figures 2 and 7.

11. Page 8, lines 4-5. Should a reference or references for NPF be added?
-Added references:

Dal Maso, M., Kulmala, M., Riipinen, I., Wagner, R., Hussein, T., Aalto, P. P., and Lehtinen, K. E.: Formation and growth of fresh atmospheric aerosols: eight years of aerosol size distribution data from SMEAR II, Hyytiala, Finland. Boreal Environment Research, 10(5), 323, 2005.

Kulmala, M., Kontkanen, J., Junninen, H., Lehtipalo, K., Manninen, H.E., Nieminen, T., Petäjä, T., Sipilä, M., Schobesberger, S., Rantala, P. and Franchin, A. et al.: Direct observations of atmospheric aerosol nucleation. Science, 339(6122), pp.943-946, 2013.

Tunved, P., Hansson, H. C., Kerminen, V. M., Ström, J., Dal Maso, M., Lihavainen, H., Y. Viisanen, Y., Aalto, P.P., Komppula, M. and Kulmala, M.: High natural aerosol loading over boreal forests. Science, 312(5771), 261-263, 2006.

12. Page 8, lines 20-21. I can see how this sentence is needed, but it seems to be just tacked onto the end of the section.
-This sentence is removed, because only aerosol concentrations higher than 0.1 cm$^{-3}$ are shown now on the figure. (This sentence is not needed anymore as it explained large variability in the concentrations below that).

13. Page 10, lines 9-10. How would small scale mixing lead to the behavior that is shown in the figure?
-We suspect that this is a response of aerosol growing rapidly as it moves from very dry air to much moister conditions, supported by the lowering of the depolarization ratio in the same region. This requires some mixing over small vertical length scales between two otherwise stable layers, otherwise such a signal would be rapidly 'smeared out'. Additional evidence is required to confirm this hypothesis.
The sentence is rephrased: This thin layer could be either a result of limited small-scale mixing between two layers, that were probably stable, or the result of large-scale transport of smoke or dust; however, we suspect that this is a response of aerosol growing rapidly as it moves from very dry air to much moister conditions, especially since the low HSRL circular depolarisation values suggest more spherical particles in this thin layer.

14. Page 11, line 10. Suggest using the same units as shown on the figure.
-We have changed the units as suggested.

15. Page 12, line 2. The text mentions deep convection, but can that be safely said from the data that has been presented so far? Wouldn't the HSRL have issues seeing the cloud-top height?
-The HSRL does not see the cloud top heights (there is complete attenuation of the signal where there are clouds), but the cloud-top height was seen in the cloud radar that operated during the BAECC campaign.
We do not intend to add a figure showing the radar reflectivity so we have removed 'deep'.

---

## Author Comment (AC3) · 3 May 2018

Review of : Combining airbone in-situ and ground based Lidar measurements for attribution for attribution of aerosol layers.
By Nikandrova et al.
The authors describe two case studies (clear sky and cloudy sky) observed over the SMEAR-II station during a field campaign in 2014. The authors used airborne measurements (mostly in-situ size distributions) associated to ground based HSRL Lidar.
This manuscript is of interest for the scientific community but need major revisions before submission to ACPD.

**Response to comments from Anonymous Referee #4**

We thank the referee for the constructive comments to help us to improve the manuscript. Below please find our answers to the comments.

Fist of all, the aim of the paper is pretty vague: ' investigate aerosol layers in a rural environment' and need to be clarified.
We have tried to clarify the aim of the paper in the introduction and added several sentences: 'We were particularly interested in how the aerosol size distribution varied both within and between layers. This information could be used to determine whether there was mixing within and between layers, and whether there had been any recent contact with the surface.'
We have also added this sentence at the end of the last paragraph of the introduction: 'Back trajectory analysis was conducted for both case studies to examine whether these analyses produced similar layer structures to those observed, and how closely the diagnosed layer altitudes corresponded with those observed by the HSRL.'

This paper is showing size distribution differences that occur within each layer of the atmosphere as a function of time. The authors interpret each increase of the fine particle number concentration as a nucleation event within each layer. However, the differences of Aitken, Accumulation and Coarse number concentrations are only pointed out.
Although we interpret these increases as NPF, we focus on how variable the aerosol size distribution is within each layer, and how the shape of the aerosol size distribution changes over time; this can then be used to infer any mixing within or between layers. The relative lack of mixing observed in elevated layers may then inform likely evolution of aerosol undergoing long-range transport.

The conclusions of this paper needs some work : The comparison of the RH and HSRL profiles with HYSPLITT results are most of the times in good agreement but the ''heights did not always coincide''. These height differences are not expressed in the main part of the paper and should probably be. . .
We have added more text to the main part after this sentence 'During 9 and 10 April, for example, the trajectory BL height was lower than the BL seen from the HSRL and consequently, the trajectory analysis suggested a thicker middle layer.', 'BL height diagnosed from trajectory analyses was 50-800 m lower than that observed, whereas for elevated layers, the layer boundary heights were better represented, with departures typically less than 200 m. These larger height differences for layers associated with the BL top are attributed to the difficulties that meteorological models have in representing the BL (e.g. Holtslag et al., 2013), which are then propagated through to the trajectories.'

Added to the conclusion: Errors in trajectories (particularly in the vertical) arise from the difficulties that the meteorological models providing the wind fields have in accurately representing vertical motion and turbulence, the boundary layer, and other sub-grid scale features (Stohl et al., 2001, Riddle et al., 2006, Hoffmann et al., 2016). Uncertainties in the horizontal can be determined using ensemble trajectory techniques (Stohl et al., 2001) but these are less likely to capture vertical discrepancies arising from processes that the meteorological model may not capture correctly, such as the boundary layer.

L. Hoffmann, T. Rößler, S. Griessbach, Y. Heng and O. Stein, Lagrangian transport simulations of volcanic sulfur dioxide emissions: Impact of meteorological data products, *Journal of Geophysical Research: Atmospheres*, **121**, 9, (4651-4673), (2016).

Holtslag, A.A., G. Svensson, P. Baas, S. Basu, B. Beare, A.C. Beljaars, F.C. Bosveld, J. Cuxart, J. Lindvall, G.J. Steeneveld, M. Tjernström, and B.J. Van De Wiel, 2013: Stable Atmospheric Boundary Layers and Diurnal Cycles: Challenges for Weather and Climate Models. Bull. Amer. Meteor. Soc., 94, 1691–1706, https://doi.org/10.1175/BAMS-D-11-00187.1

Riddle, E. E., P. B. Voss, A. Stohl, D. Holcomb, D. Maczka, K. Washburn, and R. W. Talbot (2006), Trajectory model validation using newly developed altitude‐controlled balloons during the International Consortium for Atmospheric Research on Transport and Transformations 2004 campaign, *J. Geophys. Res.*, 111, D23S57, doi:10.1029/2006JD007456.

Stohl, A., L. Haimberger, M. P. Scheele, and H. Wernli. "An intercomparison of results from three trajectory models." *Meteorological Applications* 8, no. 2 (2001): 127-135.

A brief presentation of the Hysplitt model and especially the resolution of the data input of the model could help the authors to interpret these differences.
Added to the text: The National Center for Environmental Prediction (NCEP) Global Data Assimilation System (GDAS) dataset with one degree resolution was used for the meteorological input to the model.

Also the last conclusion of the paper is that the synergy between radiosounding, LIDAR and back trajectories gives more confidence in determining the air mass origin. Is this really the main conclusion ?
This is one of the main conclusions, and we have added more text (also written in the comment above: Errors in trajectories (particularly in the vertical) arise from the difficulties that the meteorological models providing the wind fields have in accurately representing vertical motion and turbulence, the boundary layer, and other sub-grid scale features (Stohl et al., 2001, Riddle et al., 2006, Hoffmann et al., 2016). Uncertainties in the horizontal can be determined using ensemble trajectory techniques (Stohl et al., 2001) but these are less likely to capture vertical discrepancies arising from processes that the meteorological model may not capture correctly, such as the boundary layer.)

Last, the authors state: 'Evidence for cloud processing of aerosol particles was also seen in the BL but the amount of processing varied [. . .]'. The authors are showing Hoppel minimum that could be related to cloud processing but it's not supported by real evidence. It could also be due to different sources of aerosol with one source quite close to the instrumental site ? Moreover, I believe you can't talk about the 'amount of processing'. . .
We agree that we have no direct evidence of cloud processing. However, the lower part of the BL was very well-mixed which suggests that any local sources should also be reasonably well-mixed; there is low variation in other size ranges for the entire BL.

This sentence has been reworded: 'In the BL, the aerosol size distribution displayed a Hoppel minimum suggesting cloud processing of aerosol particles, but with variations that were presumably again due to the specific nature of the updrafts and downdrafts resulting in BL mixing that was not fully homogeneous in the upper part of the BL.'

Minor corrections:

P3 L 4 : Not well said. Please rephrase
Rephrased: 'Microphysical properties retrieved from HSRL-2 showed a good agreement with in situ measurements; however, backscatter and extinction coefficients calculated from corresponding in situ measurements were consistently underestimated, which was attributed to the undersampling of coarse mode particles by in situ measurements'

Page 3 L15 : Needs to add references to support that like Crumeyrolle et al., 2010; Rose et al., 2015a, Berland et al., 2016.
Rephrased and added two of suggested references: This suggests that, in the boreal forest, large-scale NPF events are typically confined to the BL, similar to results found in other environments (Crumeyrolle et al., 2010; Berland et al., 2016).
Rose et al., 2015a reported NPF events in the free troposphere over Mediterranean.

Figure 2 : You are always refering to the mode you define P5. Could you add on your size distribution plots the limits of each mode (nucleation, Aitken, Accumulation, Coarse). It would help the reader.
Added as suggested

No error bars on the Figure 2i within the small particles range for the middle layer ?
There is no error bar because shaded areas show variability in the layers, and in the middle layer during this flight, small particles were detected only once. This is explained in the text on p. .

P10 L7 : 'A very high peak' : could you add in comparison to the rest of the profile ?
Added as suggested

P10 L10 : Smoke or Dust are not known to be spherical particles …
A clause was missing from our sentence. The sentence has been rephrased 'This thin layer could be either a result of limited small-scale mixing between two layers, that were probably stable, or the result of large-scale transport of smoke or dust; however, we suspect that this is a response of aerosol growing rapidly as it moves from very dry air to much moister conditions, especially since the low HSRL circular depolarization values suggest that particles in this thin layer were relatively spherical.'

P11 L14 : below 100nm instead of 30nm
We left 30 nm as originally written because on the figure 7c aerosol number size distribution in the upper layer (blue) lower than in the first middle layer (yellow) below 30 nm.

P11 L 22 : Please add 100nm to show the reader where the Hoppel minimium is located.
Added as suggested

P12 L17 : around 100nm replace with around 70nm
Replaced as suggested

Figure 7. Not able to distinguish the 3 green shades…
We have changed colours.

Figure 9 : No error bars : Does it mean that you used only one spectra. If yes than it needs to be stated somewhere.
Error bars added to the plot.

5 L25 : Please add explanations. I don't want to read Laakso et al. To understand what you did. The GF is usually dependant of the different compounds present in the aerosol. So how did you get this information ?
Added to the text: 'using a growth factor (GF) calculated for a boreal forest environment using measurements from Hyytiälä station by Laakso et al. (2004). They weighted the GF for compounds with different hygroscopicity according to their respective fractions to obtain an optimal combined GF coefficient.'

P7 L25 : Hard to tell cause there are no measurements of the fine particle number concentration within the middle layer…
No particles smaller than 15 nm were detected in the upper and middle layers during the morning flight even though the detection limit is 10 nm, but these were observed in the afternoon flight, providing evidence for NPF in elevated layers.

P9 L27-29 : Please tell us more about the difference you see cause it's not obvious for me.
This paragraph is rephrased to be clearer: 'For particles smaller than 300 nm, the shape of the size distribution and the number concentrations changed from day to day. For particles larger than 300 nm, while the number concentration varied, the shape of the distribution remained similar across all 3 days.'

P11 L 20 : If you are implying that the cloud base is playing a role in the mixing efficiency be more clearer
We have rephrased this sentence 'Two tendencies are seen in the BL: a more mixed lower part up to about 1000 m where the cloud bases were, and a less mixed upper part' as we did not mean to imply whether cloud base plays a role. It happens that saturation occurs at a similar altitude as the mixing profile begins to depart from a well-mixed profile, but we do not infer that cloud is necessarily the cause of the change in the mixing profile.
Our sentence now reads 'The BL was well-mixed up to 600 m, and became progressively less well mixed above this, with convectively buoyant air parcels reaching up to 2500 m. The radiosonde thermodynamic profile suggested that deep convection to 4 km or so was possible, and did indeed occur later on in the day.'

P12 L 19 : Any interpretation why there is less particles above 500nm ?
Added to the text: This may due to dilution as the growing BL entrains air from above with lower concentrations in this size range.

P12 L22 : Do you mean that nucleation occurs over the cloud top ? Please add references to support this.
With cloud-driven entrainment we are only implying that there might be localized mixing, rather than a fully-mixed layer. Hence, there may be pockets with slightly higher and slightly lower concentrations, without NPF necessarily occurring.

Section 3.1.2 : If you are talking about erors you need to state the number of SD you used to get the average showed in figure 2. ..
We report uncertainties using one standard deviation, as written in the caption for the figures and in the text.

---

## Author Comment (AC5) · 4 May 2018

This is a nice workup of case studies using multiple sources of data (lidar profile measurements, relative humidity from radiosondes, in situ size distributions, and backtrajectory analysis). Although it is somewhat limited in scope, I think the analysis successfully uses these multiple disparate data sources to gain a deeper understanding of the atmospheric layers in the case studies. The figures are informative and well constructed for showing correspondence between different measurement types and for illustrating interesting aspects of the case studies. I recommend publication after addressing a few points.

**Response to comments from Anonymous Referee #1**

We thank the referee for the constructive comments to help us to improve the manuscript. Below please find our answers to the comments.

Specific comments:

Page 2, line 30. Delete "at higher latitudes". Smoke aerosol is not limited to high latitudes.
-Deleted as suggested.

Page 4, line 13. "the cross-polarization channel measures the degree of circular polarization". I think this should probably be reworded. I don't think just one channel by itself can measure the degree of polarization; it must be compared to another channel.
-Rephrased: the cross-polarization channel measures the degree of circular polarization relative to the combined channel.
A related question: what is the polarization state of the combined channel? That is, does the polarization split occur before or after the Rayleigh-Mie split?
- The polarization split occurs before the Rayleigh-Mie split.

Page 4, line 14. I would have liked to look up the answer to my previous question in the quoted reference (Goldsmith 2016) but it isn't in the bibliography.
-Added to the bibliography

Page 4, line 24. What is the particle size cut off of the inlet?
-The aerodynamic particle cut off diameter is 5.0 um. (McNaughton, 2007)
Aerosol size distributions in the figures 2,4,7, 9 are now shown until 5 um.

Page 5, line 29-31. Are these quoted sizes radius or diameter?
-Added 'with diameters'

Page 10, line 11. "aged dust, especially since the low HSRL circular depolarization values suggest more spherical particles". I am confused by this sentence. Dust, even aged dust, would be expected to be dominated by non-spherical particles. Either I'm misunderstanding the intent of the sentence (in which case, please reword) or else you are suggesting that aged dust would be expected to have spherical depolarization values similar to what's observed. If that's the intent, please include more discussion and references to support this idea.
-We suspect that this is a response of aerosol growing rapidly as it moves from very dry air to much moister conditions, supported by the lowering of the depolarization ratio in the same region. This requires some mixing over small vertical length scales between two otherwise stable layers,

otherwise such a signal would be rapidly 'smeared out'. Additional evidence is required to confirm this hypothesis.

A clause was missing from our sentence. The sentence has been rephrased 'This thin layer could be either a result of limited small-scale mixing between two layers, that were probably stable, or the result of large-scale transport of smoke or dust; however, we suspect that this is a response of aerosol growing rapidly as it moves from very dry air to much moister conditions, especially since the low HSRL circular depolarization values suggests that particles in this thin layer were relatively spherical'

Figures 1 seems to show enhanced depolarization during the time period selected for the case study (8 April). Any comment about what this might indicate?

-It might be long-range transport of pollution, that is already discussed on p. 7:

'The second middle layer had a similar size distribution shape for particles smaller than 100 nm but higher concentrations, and displayed the highest concentrations of supermicron particles, even higher than in the BL. The second middle layer also exhibited much more depolarization than the other layers (Fig. 1b), together implying long-range transport of large non-spherical particles'

Lidar ratio can give important insight into aerosol type and therefore would potentially provide another useful clue for analyzing the case studies. Also, there is significant interest in the aerosol lidar community in cataloging lidar ratio for different aerosol scenarios. HSRL measures backscatter and extinction separately and therefore includes lidar ratio. Why not include lidar ratio in Figures 1 and 6 and in the analysis?

- Lidar ratio was outside the scope of this work, but it will be provided to the community in the next papers including data from the whole BAECC campaign, not only from our case studies.

Page 10, line 23 discusses the depth of cumulus clouds. Since these block the laser light, it's not clear how you estimate the top-heights of these clouds. Please explain.

-The cloud-top height was seen in the cloud radar that operated during the BAECC campaign. The sentence is rephrased: The cloud radar showed that occasional cumulus clouds were formed from 1000 m in altitude and were able to grow to at least 3000 m in altitude by late afternoon.

In the discussion section, please include more discussion of the proposed mechanisms for new particle formation in the particular cases discussed. I realize there are no measurements available to explain this definitively, but I think some more specific discussion of possibilities supported by literature references would be helpful. Specifically, you discuss new particle formation in the boundary layer for case 1 and then use back-trajectory analysis to infer that the airmass originated over the Arctic Ocean.

Does this mean that the new particle formation occurred over the Arctic Ocean? Was this area covered by sea ice? You also suggest that new particle formation occurred in the elevated layer at the same time. What are published mechanisms for new particle production over sea ice and in elevated layers that would be consistent with these observations?

- The NPF described in the manuscript happens in the boreal forest. Air masses coming from the Arctic Ocean (clean area) are known to be good for NPF in Hyytiälä. Tunved et al. (2006) shows not only that NPF in Hyttiälä is preferred in originally clean marine air masses, but that the NPF is initiated soon after this air enters the boreal forest zone.

Tunved, P., Hansson, H. C., Kerminen, V. M., Ström, J., Dal Maso, M., Lihavainen, H., Y. Viisanen, Y., Aalto, P.P., Komppula, M. and Kulmala, M.: High natural aerosol loading over boreal forests. Science, 312(5771), 261-263, 2006.

Other references describing NPF at the station have been added:

Dal Maso, M., Kulmala, M., Riipinen, I., Wagner, R., Hussein, T., Aalto, P. P., and Lehtinen, K. E.: Formation and growth of fresh atmospheric aerosols: eight years of aerosol size distribution data from SMEAR II, Hyytiala, Finland. Boreal Environment Research, 10(5), 323, 2005.

Kulmala, M., Kontkanen, J., Junninen, H., Lehtipalo, K., Manninen, H.E., Nieminen, T., Petäjä, T., Sipilä, M., Schobesberger, S., Rantala, P. and Franchin, A. et al.: Direct observations of atmospheric aerosol nucleation. Science, 339(6122), pp.943-946, 2013.

Typos, etc.

Page 4, line 14. "Goldsmith" misspelled

-Changed

Page 4, line 24. Is this liters per minute? Can the "L" be capitalized? It looks like a "one".

-Changed to L min$^{-1}$

Page 5, line 14. "for the algorithm" is not clear. Do you mean for the layer-detection algorithm?

-Added as suggested

Page 5, line 18. "most often indicate edges of layers". Fragmented sentence.

-Rephrased: Layers classified with the HSRL were confirmed with the RS measurements, where edges of layers could be seen in changes of specific and relative humidity profiles.

Page 7, line 1. "this layer" is not clear, since you mention four layers. Which layer?

-Changed 'this layer' to 'the BL'.

Table 1. Please explain acronyms in the table caption (particularly "NPF").

-Changed: Acronyms are explained in the table caption.

Also, the formatting of the "MidLII" column is strange in that it is unlike any other column in having both the height and depth. I realize this is to save space since there is only one layer. Another possibility that might be clearer is removing the "MidLII" column and putting two sets of measurements (separated by a comma) in that row of the "MidL height" and "MidL depth" columns.

-Changed as suggested.

Figures 2 and 7, the annotations are hard to read. Repeating the information from the color legend in the caption would help. It would also be useful to indicate the layer boundaries as lines or markers on the humidity profile or lidar curtain so that it would be more immediately obvious where the in situ size distributions are applicable.

- Boundaries are added to the RH plot on fig. 2 and 7 as suggested, and legend is changed, so it can be easier to read.

Also, it would be useful to make the axis labels bigger in Figures 2, 3, 7, 8 and 9.

- Figure axes are already as big as possible to fit text nicely.

There seems to be a rendering or smoothing artifact in the lidar curtain in Figure 2e that shows as a series of horizontal lines where the lidar backscatter profile does not change for 15 or 20 minutes between 11:50 and 12:10.

-Smoothing artefact due to MATLAB plotting issue in data gaps - Fig. 2e now corrected. Data gap was due to calibration period.

---

## Author Response (AR2)

**Editor:**

-Reply to reviewer 1: Please add the information on the Rayleigh-Mie split and the size cut of the inlet to the manuscript.

Added (page 4 line 21 and line 32)

- Add "e.g." in the citation parenthesis when referencing with only a limited and selective amount of literature (e.g. on the health effects on page 1 or on the general aerosol-cloud discussion on page 2).

Added

- Page 2, line 18 -19: Add relevant reference.

Added:

Clarke, A. D., Collins, W. G., Rasch, P. J., Kapustin, V. N., Moore, K., Howell, S., and Fuelberg, H. E.: Dust and pollution transport on global scales: Aerosol measurements and model predictions, J. Geophys. Res., 106(D23), 32555–32569, doi: 10.1029/2000JD900842, 2001.

- Page 3, line 13: Please fix the correct reference here.

They were added on page 3, line 17-18 already before, just the comment remained in a wrong place

- Page 3, line 27: Here and throughout the manuscript, please use the correct abbreviations (e.g. "Sect.", see https://www.atmospheric-chemistry-and-physics.net/for_authors/manuscript_preparation.html)

Corrected for Sect. and Fig.

- Page 4, line 3: Add the degree and minute signs to the geographic location.

Added

- Page 4, line 29: Remove the dot after "see".

Removed

- Page 5, line 8: Please precise if this is relative or absolute accuracy in RH.

Information about the sensor that was used on the plane is now deleted, because it was a bit slow and not used for the RH corrections. Radiosonde profiles of RH are now used for calculation of the growth factor. Most of the figures did not change due to this, but there is a small change in figure 7c with a bit shifted distribution in the boundary layer. This change in the figure does not affect any conclusions.

- Page 5, line 12: Remove comma before 2014.

Removed

- Page 5, last sentences: The correction with respect to particle hygroscopicity is not really clear. Please specify, if you have used one growth factor distribution (as a function of RH and dry diameter) for the entire campaign or if not, what exactly you have done. If you have used one fixed GF-parameterization, I would also recommend to mention GF at RH=90% for one example dry diameter. This may facilitate further interpretation.

One GF (from Laakso et al. 2004) was used and implemented as a function of RH.

The process how the Laakso et al. calculated the GF is now deleted and one example added, so now the text reads: 'The GF was implemented as a function of RH for the SMPS data and was between about 1.2 and 1.4 at 90 % RH for the range of the observed particle sizes (20-230 nm).'

- Page 7, line 7: "Fig." (see comment above).

Changed

- Page 10, Line 15: Please rephrase the last sentence, stating that the upper layer may have been influenced from mixing with layers above the maximum altitude.

Rephrased as suggested

- Page 11, line 17: Please reference correctly to the corresponding figure or remove the sentence if not needed.

Changed

- The conclusion needs some further refinement. Please remove repetitions (especially technical information) which are already mentioned in the sections before (e.g. that trajectories were calculated every 50 m or the technical details in the first paragraph) and focus on the main findings.

I would also recommend to rephrase the last sentence of your conclusions since the current sentence states a very obvious fact (and not a new finding) that more information will give more confidence in the determination of air mass origin. As it currently reads, this last sentence could be interpreted as your main finding.

Repetition is deleted and the text rephrased

**Referee 3:**

Minor comments.

Page 4, line 31. What is the cutoff size associated with the inlet? Is the inlet isokinetic?

The aerodynamic particle cut off diameter is 5.0 um. (McNaughton, 2007) – added to the text Inlet is isokinetic under the airspeed that we used.

Page 5, line 7. In their response to the reviewer comments the authors highlight that the RH is generally less than 50%. Could that be included in the text in this section?

Added on page 6 line 4

Page 6, line 23. I am still bothered by the lack of a scale for the RH in these Figure 1. I agree that it is easy to see the relative changes, but it is difficult to determine how important they are with no scale on the figure. I encourage the authors to consider ways to add an RH scale to the figure.

The absolute values are not important for this figure, because the idea is to show that change in the relative humidity coincide with the borders of the layers. And on the figure scales would not be seen well, because there are so many profiles. RH profiles closest to the flights can be clearly seen on Fig. 2 and 7.

Page 7, line 5. Should the phrase "in the zone" be added after "backscatter values"?

Added as suggested

Page 8, line 20. There are some size distributions in Figure 2 that have very small (or near zero) standard deviations. One example is the nucleation mode particles in the middle layer (2i). Is there an explanation for the small amount of variability?

There is an explanation on page 8 line 30: 'It is difficult to ascertain the variability in the nucleation mode size range ($< 25$ nm) for the elevated layers, as there may have been no particles, or too few for the instrument to obtain reliable counts.'

**Referee 1:**

In their response, the authors gave a helpful explanation of the new particle formation described in the manuscript, which was much more specific and clearer to me than the manuscript. I wish this explanation (or a version of it) could also be included in the manuscript. I refer to this response: "The NPF descried in the manuscript happens in the boreal forest. Air masses coming from the Arctic Ocean (clean area) are known to be good for NPF in Hyytiala. Tunved et al. (2006) shows not only that NPF in Hyttiala is preferred in originally clean marine air masses, but that the NPF is initiated soon after this air enters the boreal forest zone."

Added to the Sect. 3.1.2

[revised manuscript text omitted]

---

## Author Response (AR3)

We thank the editor for the minor comments to improve our figures. We have done all the things he has suggested.

I agree with reviewer #3 that a scale info of the RH profiles should be added to Fig. 1. It is not easily clear if these are large RH changes or not. Maybe add a small scale to the upper part of the figure or add (exemplarily) min/max-values.

Scale is added from 0 to 100.

The last comment of reviewer #3 raises the question if you have enough scans to actually calculate mean size distributions. If it is only one scan, it might not hold the argumentation made. Maybe mark them with a different color (e.g. grey) and discuss that the interpretation for this size range is limited due to insufficient data.

Figure is changed.
Text added: "Due to relatively low resolution of the SMPS measurements (2 minutes and 300 m) nucleation mode particles were detected only at one height in the middle layer. Therefore, there were not enough measurements to calculate mean size distribution and standard deviation, and these measurements are shown in grey (Fig. 2i). Nevertheless, we believe that the measured nucleation mode particles in this layer are not an artefact, because increased concentration of particles was also observed by the uCPC at the same heights as theSMPS (not shown)."

[Figure]

One last comment on the figures in general: Please make sure that the font size of the figures is approx. equal to the font size of the caption. Some of the figures (e.g. 3, 4, 9) have really font sizes and will be difficult to read in the printed version later on.

Sizes are increased

[revised manuscript text omitted]